# Quantitative Proteomics and Differential Protein Abundance Analysis after Depletion of Putative mRNA Receptors in the ER Membrane of Human Cells Identifies Novel Aspects of mRNA Targeting to the ER

**DOI:** 10.3390/molecules26123591

**Published:** 2021-06-11

**Authors:** Pratiti Bhadra, Stefan Schorr, Monika Lerner, Duy Nguyen, Johanna Dudek, Friedrich Förster, Volkhard Helms, Sven Lang, Richard Zimmermann

**Affiliations:** 1Center for Bioinformatics, Saarland Informatics Campus, Saarland University, 66041 Saarbrücken, Germany; pratiti.bhadra@bioinformatik.uni-saarland.de (P.B.); duy.nguyen@dkfz-heidelberg.de (D.N.); volkhard.helms@bioinformatik.uni-saarland.de (V.H.); 2Medical Biochemistry and Molecular Biology, Saarland University, 66421 Homburg, Germany; stefan.schorr@uks.eu (S.S.); Monika.Lerner@uks.eu (M.L.); Johanna.Dudek@uks.eu (J.D.); sven.lang@uni-saarland.de (S.L.); 3Bijvoet Center for Biomolecular Research, Utrecht University, 3584 CH Utrecht, The Netherlands; f.g.forster@uu.nl

**Keywords:** endoplasmic reticulum, mRNA targeting, protein targeting, protein import, membrane protein insertion, protein translocation, Sec61 complex, TIGER domain, label-free quantitative mass spectrometry, differential protein abundance analysis

## Abstract

In human cells, one-third of all polypeptides enter the secretory pathway at the endoplasmic reticulum (ER). The specificity and efficiency of this process are guaranteed by targeting of mRNAs and/or polypeptides to the ER membrane. Cytosolic SRP and its receptor in the ER membrane facilitate the cotranslational targeting of most ribosome-nascent precursor polypeptide chain (RNC) complexes together with the respective mRNAs to the Sec61 complex in the ER membrane. Alternatively, fully synthesized precursor polypeptides are targeted to the ER membrane post-translationally by either the TRC, SND, or PEX19/3 pathway. Furthermore, there is targeting of mRNAs to the ER membrane, which does not involve SRP but involves mRNA- or RNC-binding proteins on the ER surface, such as RRBP1 or KTN1. Traditionally, the targeting reactions were studied in cell-free or cellular assays, which focus on a single precursor polypeptide and allow the conclusion of whether a certain precursor can use a certain pathway. Recently, cellular approaches such as proximity-based ribosome profiling or quantitative proteomics were employed to address the question of which precursors use certain pathways under physiological conditions. Here, we combined siRNA-mediated depletion of putative mRNA receptors in HeLa cells with label-free quantitative proteomics and differential protein abundance analysis to characterize RRBP1- or KTN1-involving precursors and to identify possible genetic interactions between the various targeting pathways. Furthermore, we discuss the possible implications on the so-called TIGER domains and critically discuss the pros and cons of this experimental approach.

## 1. Introduction

Protein import into the ER is the first step in the biogenesis of about 10,000 different soluble and membrane proteins of nucleated human cells, representing approximately one-third of the proteome [1,2,3,4,5,6]. These proteins fulfill their functions in the membrane or lumen of the ER plus the nuclear envelope, in one of the organelles belonging to the pathways of endo- and exocytosis (i.e., ERGIC, Golgi apparatus, endosome, lysosome), in lipid droplets, or at the cell surface as plasma membrane or secretory proteins. Typically, ER protein import involves membrane targeting as the initial step, and the insertion of nascent membrane proteins into or translocation of soluble precursor polypeptides across the ER membrane as the second step (Figure 1). Both processes depend on N-terminal signal peptides (SPs) or N-terminal signal transmembrane helices (TMHs), which serve as targeting signals, in the precursor polypeptides [7,8,9,10,11]. The Sec61 complex of the ER membrane represents the entry point for most precursor polypeptides [12,13,14,15,16,17,18,19,20,21,22,23,24,25]. However, some precursors of membrane proteins are handled by dedicated membrane protein insertases such as ER membrane protein complex (EMC) and TMCO1 complex [26,27,28,29,30,31,32,33]. 

Cryo-electron tomography shows the heterotrimeric Sec61 as a large multicomponent ensemble together with ribosomes and the membrane-embedded translocon-associated protein (TRAP) complex and oligosaccharyltransferase (OST), the multi-subunit enzyme that catalyzes N-linked glycosylation [18,20,21,23]. This super-complex or Sec61 translocon can insert into the membrane or import into the lumen a whole variety of different precursor polypeptides (soluble proteins, type I-, type II-, type III-, tail-anchored- (TA-), and hairpin-membrane proteins). Subsequently, these precursors mature to membrane proteins with one, two, or multiple transmembrane helices, glycosylphosphatidylinositol- (GPI-) anchored membrane proteins, or soluble proteins in the ER lumen, such as ER-resident proteins or secretory proteins [5,6]. Membrane insertion and import are either mediated by a cleavable N-terminal SP or the N-terminal TMH of the nascent precursor polypeptide. Cleavable SPs are removed from the inserting or incoming precursor polypeptides by the signal peptidase complex (SPC) [34,35,36]. 

Prior to ER entry, precursor polypeptides have to be targeted to the ER membrane [4,5,6]. Cotranslational ER targeting is mediated by the cytosolic signal recognition particle (SRP) and its heterodimeric receptor in the ER membrane, the SRP receptor (Table 1) [37,38,39,40,41,42,43,44,45,46,47]. Since ribosome-nascent chain (RNC) complexes are targeted in this case, this mechanism also represents an mRNA-targeting mechanism, here driven by an N-terminal SP or TMH and not directly by the mRNA (Figure 1, upper part). In post-translational import via the Sec61 complex, ER targeting occurs via a similar binary system of cytosolic component (SND1) and its heterodimeric receptor in the ER membrane (SND2 plus SND3) or via direct contact with the Sec61 complex and any of its associated components, such as Sec62 [48,49,50,51,52,53]. The SRP-independent targeting pathway to the Sec61 complex was first identified in yeast and named the SND pathway. In human cells this targeting pathway can be used by small presecretory proteins (i.e., precursors with less than 100 amino acid residues) [50,51,52]. In addition to the above-mentioned membrane proteins, the ER membrane also contains hairpin- and TA-membrane proteins, which depend on dedicated components and post-translational pathways for their ER targeting and membrane insertion (Figure 1, upper part). The TRC pathway handles TA proteins [54,55,56,57,58,59,60,61,62] and the PEX3-dependent pathway certain hairpin proteins, which remain in the ER or, eventually, end up in lipid droplets [63,64]. In the case of the TRC and PEX pathways, targeting to these membrane components is mediated by the Bag6 complex plus additional cytosolic factors and PEX19, respectively. One lesson from the analysis of these pathways is that there are some precursor polypeptides that can be targeted to the ER by more than one pathway. Some small human presecretory proteins can be targeted to the Sec61 complex via the SRP, SND, and TRC pathways or directly via Sec62 [50,51,52,53]. Likewise, some tail-anchored membrane proteins, such as Sec61ß, can be targeted to the membrane via the same three pathways as small presecretory proteins [49]. Thus, the targeting pathways have overlapping substrate specificities and can substitute for each other to a certain extent. Notably, the Sec61ß coding mRNA can also be targeted to the ER by an unknown mechanism [65]. Last but not least, some of the precursors of mitochondrial proteins are, apparently, targeted to the ER and, subsequently, moved from the ER to mitochondria via a process that was named ER-SURF and identified in yeast [66].

There is clear evidence that for Mammalia, yeast, and plants there also is SP- and TMH-independent and, therefore, SRP-independent targeting of mRNAs to the ER (Figure 1, lower part) [67,68,69,70,71,72,73,74,75,76,77,78,79,80,81,82,83,84,85,86,87,88,89,90,91,92,93,94,95,96]. According to original biochemical and cell-biological analysis in Mammalia, the synthesis of various polypeptides, including cytosolic proteins, is initiated on 80S ribosomes or 60S ribosomal subunits that are continuously attached to the ER membrane [72,73,74]. Here, mRNA targeting is supposed to be mediated by putative mRNA receptors in the ER membrane, such as AEG-1 (also termed LYRIC, MTDH) [90], LRRC59 (also termed p34) [68,70,95,96], RRBP1 (also termed p180) [67,69,80,81,82,84,85], and KTN1 1 (kinectin 1) [75,77] in Mammalia (Table 1). Notably, these putative receptors are not conserved in plants and lower eukaryotes [76,79,93,94]. According to proximity-specific ribosome-profiling experiments, however, ER targeting of RNCs with nascent polypeptide chains, which are not yet long enough to be able to interact with SRP, play a more dominant role in mRNA targeting to the ER than direct mRNA targeting to permanently ER-associated 80S ribosomes or 60S ribosomal subunits [87]. By definition, the first is translation-dependent, the latter independent (Figure 1, lower part). Either way, broad knowledge about the possible specificity of these mRNA-targeting reactions is only beginning to accumulate [86,91]. Until recently, there were only a few examples for precursor polypeptides, such as the SP-comprising precursors of the GPI-anchored plasma membrane protein placental alkaline phosphatase [84,85], of certain Collagens [81], and of the ER-resident protein calreticulin [84] that are known to involve RRBP1 either as a receptor for RNCs or mRNA. Notably, as mentioned above, the precursor of the TA membrane protein Sec61ß has been found to involve RRBP1-independent mRNA targeting [65]. Most recent data from mRNA crosslinking or ribosome proximity labeling and accompanying transcriptome analysis, however, gave a first global glimpse of the possible full client spectra of two mRNA receptors, namely AEG-1 [90] and LRRC59 [96].

We also started to globally identify precursor polypeptides, which involve targeting of the corresponding mRNAs in their ER import, irrespective of whether they are targeted directly by mRNA or by mRNA in the context of RNCs. Our approach focused on the other two putative mRNA receptors RRBP1 and KTN1 plus the ER membrane protein ERj1 because of its known abilities to interact with ribosomes as well as nucleic acids [97,98,99,100]. The approach involves gene silencing of the putative receptor of interest in three different HeLa cell cultures with two different targeting siRNAs in parallel to a non-targeting or control siRNA and differential proteomic analysis for the nine cell pools. Initially, 2D-fluorescence difference gel electrophoresis (DIGE) in combination with mass spectrometry (MS) and, subsequently, stable isotope labeling by amino acids in cell culture (SILAC) were employed. Although the number of analyzed proteins improved by changing strategies, the results remained disappointing in terms of coverage of the total cellular proteome. Therefore, the approach was switched to label-free quantitative MS analysis and differential protein abundance analysis. This protocol was previously used to characterize the client spectrum and client SP features of the Sec61 complex, TRAP complex, Sec62/Sec63 complex, translocating chain-associating membrane (TRAM) protein, i.e., another Sec61-interacting protein import component, and the ER lumenal Hsp70-type chaperone immunoglobulin heavy chain binding protein (BiP), which plays a role in both protein import of certain precursor polypeptides and in ER protein folding plus assembly [101,102,103]. The approach is based on the fact that precursors of polypeptides, which have to be imported into the ER, are degraded by the proteasome in the cytosol upon interference with their ER targeting or translocation. Therefore, their cellular levels are decreased as compared to control cells, and this change is detected by quantitative MS and subsequent differential protein abundance analysis. Typically, the decrease is accompanied by an increase of ubiquitin-conjugating enzymes. Furthermore, there may be a simultaneous increase in other ER import components, which may indicate a possible genetic interaction between different pathways. 

Here we report on negatively and positively affected proteins after depletion of RRBP1, KTN1, and ERj1 in HeLa cells and discuss the results in comparison to the previous depletions both in respect to the benefits of the experimental approach and in respect to the overlaps between the two different targeting pathways and the LRRC59- and AEG-1-dependent pathways [90,96].

## 2. Results

### 2.1. The Experimental Approach

As stated above, we set out to identify precursor polypeptide candidates that depend on RRBP1- or KTN1-dependent targeting of their mRNAs or RNCs to the ER by the combination of siRNA-mediated depletion of their cellular levels and subsequent label-free quantitative MS of the total cellular proteomes. The candidates were expected among the negatively affected proteins in subsequent differential protein abundance analysis. Previously, as a positive proof-of-concept for the approach, HeLa cells had been depleted of the Sec61 complex using two different SEC61A1-targeting siRNAs for 96 h, a time which allowed for both maximal depletion and cell viability [101]. Then, the proteomic consequences of this knock-down were assessed via label-free quantitative proteomics and differential protein abundance analysis relative to cells treated with non-targeting siRNA (Figure 2 and Figure 3). Briefly, cells (1 × 10^6^) were harvested and lysed. After digestion, peptides were purified and loaded for MS analysis [104]. Using the nanoelectrospray interface, eluting peptides were directly sprayed onto the benchtop Orbitrap mass spectrometer Q Exactive HF [105]. Raw data were processed using the MaxQuant computational platform [106]. The peak list was searched against Human Uniprot databases and proteins were quantified across samples using the label-free quantification algorithm in MaxQuant as label-free quantification (LFQ) intensities [107]. We note that LFQ intensities do not reflect true copy numbers because they depend not only on the amounts of the peptides but also on their ionization efficiencies; thus, they only served to compare abundances of the same protein in different samples [105,106,107,108,109,110]. Each MS experiment provided proteome-wide abundance data as LFQ intensities for three sample groups—one control (non-targeting siRNA treated) and two stimuli (down-regulation by two different targeting siRNAs directed against the same gene)—each having three data points (Figure 2a). Missing data points were generated by imputation, whereby two cases were distinguished. To identify which proteins were affected by Sec61α knock-down in siRNA-treated cells relative to the non-targeting (control) siRNA-treated sample, we log2-transformed the ratio between siRNA and control siRNA samples, and performed two separate unpaired *t*-tests for each siRNA against the control siRNA sample [101]. The *p* values obtained by unpaired *t*-tests were corrected for multiple testing using a permutation false discovery rate (FDR) test. Proteins with an FDR-adjusted *p* value of below 5% were considered significantly affected by knock-down of the targeted protein. The results from the two unpaired *t*-tests were then intersected for further analysis meaning that the abundance of all reported candidates was statistically significantly affected in both siRNA silencing experiments. All statistical analyses were performed using the R package SAM (https://statweb.stanford.edu/~tibs/SAM/ last accessed on 2 May 2021) [111].

Heat maps visualize clusters of proteins that showed differential abundance upon siRNA treatment. Red indicates positively affected proteins, green indicates negatively affected proteins. *SEC61A1* siRNA is referred to as rna4, *SEC61A1-UTR* siRNA as sirna5 (Figure 2b). Differentially affected proteins were characterized by the mean difference of their intensities plotted against the respective *p* values in volcano plots. Subunits of the Sec61 complex, which were depleted, and of the SRP receptor, which were up-regulated, are indicated (Figure 3a). Since the β- and γ-subunits of the Sec61 complex are TA membrane proteins, which do not involve the Sec61 complex in their membrane integration, their observed depletion was attributed to their degradation in the absence of their physiological interaction partner, the α-subunit [101]. Gene ontology (GO) terms assigned over 60% of the 482 negatively affected proteins to organelles of the endocytic and exocytic pathways, representing a strong enrichment compared to the value for the total quantified proteome (Figure 3b) [112]. In addition, significant enrichment of precursor proteins with SP, N-glycosylated proteins, and membrane proteins was detected. This suggested that the precursors of these proteins are substrates of the Sec61 complex and were indeed degraded by the proteasome upon its depletion, which was experimentally confirmed. According to bioinformatic analysis, ~30% of the total quantified proteome of roughly 7200 proteins comprises Sec61 substrates (Table 2). Thus, the experimental approach clearly underestimates the number of different precursor polypeptides that involve the Sec61 complex in their biogenesis. This was attributed to the timing of the experiment, which aimed for maximal depletion of the essential Sec61 complex in combination with minimal effects on cell growth and viability, and, therefore, neglects, e.g., proteins with long half-lives or high affinities for Sec61. The positively affected proteins included compensatory components, such as the two subunits of the SRP receptor, plus several cytosolic ubiquitin-conjugating enzymes and ubiquitin ligases, which is consistent with cytosolic accumulation of precursors and their proteasomal degradation [101]. 

Thus, the experimental strategy in human cells proved successful in analyzing the client spectrum of the Sec61 complex and the client´s characteristics under physiological conditions. However, an additional lesson was that interaction partners have also to be taken into consideration. Furthermore, these results set the stage for subsequent analyses of putative precursor-specific transport components, such as TRAP complex [101], Sec62/Sec63 complex [103], BiP [103], TRAM1 protein [102], and, now, the putative mRN-targeting components RRBP1 and KTN1. Notably, our ongoing work is directed at the co- and post-translationally operating precursor targeting components (Table 1, Figure 1) and will, together with the work on the other two mRNA receptors by C. Nicchitta´s lab [90,96], provide a first cellular glimpse on ER targeting and Sec61-dependent protein import on a global scale.

### 2.2. Quantitative Proteomic Analysis of HeLa Cells after Transient and Partial Depletion of RRBP1 by siRNA

RRBP1, which is also termed p180 for its molecular mass, comes in two isoforms, and is an ER membrane protein with a single TMH near its ER lumenal N-terminus and a large C-terminal domain in the cytosol [67,69]. Thus, by definition, RRBP1 belongs to the class of type III membrane proteins. The large cytosolic domain includes a lysine-rich region plus 54 tandem repeats of a positively charged decapeptide, both of which may provide an mRNA binding surface. Originally, it was characterized as a ribosome receptor, hence its name, but it also interacts with the cytoskeleton, in particular microtubules. The protein was implicated in mRNA and/or ribosome targeting to the ER for placental alkaline phosphatase (ALPP, a GPI-anchored plasma membrane protein), collagens Iα1 plus Iα2 and IIIγ (secretory proteins), and calreticulin (an ER lumenal protein) [81,84,85].

We applied the established experimental strategy to identify precursor polypeptides that may depend on RRBP1-dependent targeting of their mRNAs or RNCs to the ER [101]. To set the stage, two suitable *RRBP1*-targeting siRNAs were identified (Figure 4a). In the next experiment, the putative RRBP1-dependent precursors were expected among the negatively affected proteins in the label-free quantitative MS and subsequent differential protein abundance analysis. HeLa cells were treated in triplicates with two different *RRBP1*-targeting siRNAs (*RRBP1* #1 siRNA, *RRBP1* #2 siRNA) in parallel to a non-targeting (control) siRNA for 96 h. After RRBP1 depletion, 4813 different proteins were quantitatively characterized by MS, which were detected in all samples (Figure 5, Table 2, Appendix A). The MS data have been deposited to the ProteomeXchange Consortium via the PRIDE partner repository [113] with the dataset identifier: PXD011989 (http://www.proteomexchange.org last accessed on 2 May 2021). They included the expected representation of proteins with cleaved SP (6%), N-glycosylated proteins (8%), and membrane proteins (12%), which was comparable to the Sec61 experiment (Figure 5c versus Figure 3b, left large pies). Applying the established statistical analysis, we found that transient and partial RRBP1 depletion significantly affected the steady-state levels of 298 proteins: 141 negatively and 157 positively (permutation false discovery rate-adjusted *p* value < 0.05). As expected, RRBP1 itself was negatively affected (Figure 5a, volcano plots), which was confirmed by western blot (Figure 5b). Of the additional negatively affected proteins, GO terms assigned close to 37% to organelles of the endocytic and exocytic pathways (Figure 5c, large pies), which corresponds to a 1.37-fold enrichment (Figure 5c, large pies, 36.81% divided by 26.28% = 1.37) and is slightly below the average value (1.53), which was observed after depletion of BiP (1.0), Sec61 (2.37), Sec62 (0.87), Sec63 (1.76), TRAM1 (1.68), and TRAP (1.5) (Table 2 and Table 3). We also detected enrichment of proteins with SP (2.44-fold), N-glycosylated proteins (2.12-fold), and membrane proteins (1.46-fold) (Figure 5c, small pies), which was lower as compared to the Sec61 experiment (6.51, 2.83, 2.51) [38]. There was no known interaction partner of RRBP1 detected among the negatively affected proteins.

The identified precursors included 21 proteins with cleavable SP (i.e., the secretory proteins CD109, CDH13, CGREF1, COL4A2, DCD, GGH, IGFBP7, LAMB1, SEMA3C, SERPINE2, TGFBI, TNS4, the ER proteins PDIA6, SUMF1, TOR1B, and the membrane proteins CDH2, CPD, DAG1, L1CAM, PTPRZ1, SDC1), and 17 membrane proteins with TMH (not counting RRBP1), such as the two ER-resident hairpin membrane proteins ATL2 plus ATL3 and CMTR2, CYP51A1, DEGS1, FADS2, NCEH1, PARL, SOAT1, and SPTLC1 of the ER membrane, plus APMAP, CD151, DPY19L1, GPRC5A, SLC3A2, SLC38A5 of the plasma membrane plus the nuclear envelope protein SUN2 (Appendix A). Of these 38 negatively affected proteins 23 were N-glycosylated proteins. Interestingly, DEGS1 and PARL are also found in the mitochondrial membrane according to GO terms, which may suggest that these two proteins, after targeting of their mRNAs to the ER and their subsequent translation at this location, may travel from the ER to mitochondria via the newly identified ER-SURF pathway [66]. Notably, the latter may also be the case for TIMM21, which also was identified as a potential client of RRBP1.

The positively affected proteins included the cytosolic ubiquitin-conjugating enzymes MID1, NEDD4, and SH3PF1 (which had also been overproduced after BiP depletion for 72 h), consistent with the fact that there was accumulation of precursor polypeptides in the cytosol after partial RRBP1 depletion for 96 h (Appendix A). Interestingly, the proteins positively affected by RRBP1 depletion included both SRP receptor subunits (SRPRα, SRPRβ), which was confirmed by western blot for SRα (Figure 4b,c) and may indicate a genetic interaction between these two targeting pathways. We note that SRPRβ is a type I ER membrane protein with TMH and, thus, represents a protein that apparently does not depend on mRNA targeting in its membrane integration (SRPRα is a peripheral ER membrane protein and recruited to the membrane by SRPRβ). There was no indication of activation of the unfolded protein response (UPR) in the course of the 96 h knock-down, which would have been indicative of protein mis-folding in the ER and would have resulted in the overproduction of ER-resident molecular chaperones, such as BiP (coded by the *HSPA5* gene) or Grp170 (product of the *HYOU1* gene) (i.e., it would have appeared in Appendix A).

As a further and negative control for the MS results, we also carried out western blots for two proteins, which were neither negatively nor positively affected by RRBP1 depletion. The two proteins were the cytosolic 54 kilodalton subunit of SRP (SRP54) and the ER membrane protein Sec62. Both proteins were also unaffected according to western blot (Figure 4c). Taken together, the western blots confirmed the MS results for four out of four tested proteins (RRBP1, SRα, SRP54, Sec62).

### 2.3. Quantitative Proteomic Analysis of HeLa Cells after Transient and Partial Depletion of KTN1 by siRNA

KTN1, also termed kinectin 1, comes in several different isoforms, is an N-glycosylated ER membrane protein, and belongs to the kinectin protein family [75,77]. As such it binds kinesin, associates with microtubules, and may be involved in organelle and vesicle motility. Like RRBP1, KTN1 contains a single TMH near its ER lumenal N-terminus and a large C-terminal domain in the cytosol, i.e., it belongs to the class of type III membrane proteins. This large cytosolic domain includes a domain of unknown function (DUF) and a coiled-coil region. KTN1 also binds translation elongation factor delta and was suggested to be involved in the assembly of elongation factor-1 complex [75,77]. For the latter reasons, the protein was implicated in mRNA targeting to the ER. Since it is enriched in peri-nuclear sheets of the ER, it would be ideally suited to target mRNA particularly to this area of the cell.

Next, we applied the established experimental strategy to identify precursor polypeptides that may depend on KTN1-dependent targeting of their mRNAs to the ER. To set the stage, two suitable *KTN1*-targeting siRNAs were identified (Figure 4d). In the next experiment, the putative KTN1-dependent precursors were expected among the negatively affected proteins in the label-free quantitative MS and subsequent differential protein abundance analysis. HeLa cells were treated in triplicates with two different *KTN1*-targeting siRNAs (*KTN1* #3 siRNA, *KTN1* #4 siRNA) in parallel to a non-targeting (termed control) siRNA. After KTN1 depletion for 96 h, 4,947 different proteins were quantitatively characterized by MS, which were detected in all samples (Figure 6, Table 2, Appendix A). The MS data have been deposited to the ProteomeXchange Consortium via the PRIDE partner repository [113] with the dataset identifier: PXD011989 (http://www.proteomexchange.org, (accessed on 2 May 2021)). They included good representation of proteins with cleaved SP (6%), N-glycosylated proteins (8%), and membrane proteins (13%), which was comparable to the Sec61 experiment (Figure 6c versus Figure 3b, left large pies). Applying the established statistical analysis, we found that transient and partial KTN1 depletion significantly affected the steady-state levels of 70 proteins: 45 negatively and 25 positively (*p* value < 0.05). As expected, KTN1 itself was negatively affected (Figure 6a, volcano plots), which was confirmed by western blot (Figure 6b). Of the additional negatively affected proteins, GO terms assigned almost 41.5% to organelles of the endocytic and exocytic pathways (Figure 6c, large pies), which corresponds to a 1.55-fold enrichment (Figure 6c, large pies, 41.46% divided by 26.74% = 1.55) and is comparable to the average value (1.53) (Table 2). We also detected enrichment of N-glycosylated proteins (2.09-fold), and membrane proteins (1.76-fold) but not of proteins with SP (1.04-fold) (Figure 6c, small pies), which was lower as compared to the Sec61 experiment (6.51, 2.83, 2.51) [38]. Furthermore, the total number of negatively affected proteins of interest was low. The identified precursors included only three proteins with cleavable SP (secretory protein COL4A2, the ER enzyme GANAB, and the plasma membrane protein CD47), and eight membrane proteins with TMH (not counting KTN1), i.e., ATP6V0C, AVL9, BCL2L1, GALNT4, PTPLB, QPCTL, TMEM106B, of endogenous membranes and TMC1 of the plasma membrane (Appendix A). Of these eleven negatively affected proteins seven were N-glycosylated proteins. Interestingly, BCL2L1 is also found in the mitochondrial outer membrane, which reiterates that some of these proteins, after targeting of their mRNAs to the ER and their subsequent translation at this location, may travel from the ER to mitochondria via the newly identified ER-SURF pathway [66].

Strikingly, and in contrast to all previous depletions of proteins that are involved in ER protein import, KTN1 depletion affected predominantly cytosolic proteins, i.e., their level increased from 20 to 44%, representing a 2.2-fold increase (Figure 6c, large pies). A total of 21 cytosolic proteins were negatively affected, including metabolic enzymes, such as GAPDH and GAPDHS, protein kinases (i.e., OXSR1, PAK1, PDPK1, PDPK2, and ZAK), the ubiquitin-conjugating enzyme UBE2G2, and cytoskeletal components, such as junction plakoglobin (JUP), myosin 11, vinculin (VCL), and gamma-tubulin complex component 4 (TUBGCP4). This observation raises the question why cytosolic proteins should be degraded when they are synthesized on cytosolic rather than ER-bound ribosomes. Obviously, we can only speculate at this point. However, the cytoskeletal proteins which were negatively affected may point the way. For them it may be important to be produced and concentrated near their site of action rather than distributed throughout the cytosol, in particular for membrane-interacting cytoskeletal elements, such as junction plakoglobin and vinculin at adherens junctions between neighboring cells. Interestingly, this may be similarly true for certain protein kinases, such as OXSR1 that plays a role in regulating the actin cytoskeleton in response to environmental stress, PAK1 that regulates cytoskeletal reorganization for cell motility and morphology, and 3-phosphoinositide-dependent protein kinases 1 and 2 (PDPK1 and 2) that are also located at cell junctions. This clearly warrants to be tested in future work.

The positively affected proteins did not include any cytosolic ubiquitin-conjugating enzymes, arguing against significant accumulation of precursor polypeptides in the cytosol after partial KTN1 depletion for 96 h (Appendix A). Interestingly, however, the proteins positively affected by KTN1 depletion included RRBP1, which was confirmed by western blot (Figure 4e) and argues for a genetic interaction between the two putative mRNA-targeting pathways. We note that RRBP1 is a type III ER membrane protein with TMH and represents a protein that apparently does not depend on mRNA targeting in its membrane integration. There was no indication of activation of the unfolded protein response (UPR) in the course of the 96 h knock-down.

### 2.4. Quantitative Proteomic Analysis of HeLa Cells after Transient and Partial Depletion of ERj1 by siRNA

ERj1 belongs to the class of type I membrane proteins and was biochemically characterized as a ribosome-associated membrane protein (RAMP) [97,98,99,100]. Its ribosome association, however, may be more dynamic compared with the high-salt resistant RAMPs, i.e., Sec61, TRAM, and TRAP [12,13], which was biologically confirmed by fluorescence microscopy using fluorescently labeled antibodies against ERj1 in permeabilized MDCK cells [100]. According to single particle reconstruction after cryo-EM, ERj1 binds at the ribosomal tunnel exit and involves the 28S rRNA, specifically expansion segment 27 (ES27) [99]. ERj1 is supposed to play a role in ER protein import as a possible functional homolog for the Sec62/Sec63 complex, combining the cytosolic ribosome binding activity of Sec62 with the ER lumenal Hsp40-type co-chaperone activity of Sec63 in one polypeptide [98,114,115,116,117,118,119]. An involvement of ERj1 has been suggested for the precursors of serine protease inhibitors alpha1-antichymotrypsin and inter-alpha trypsin inhibitor heavy chain 4, involving the two cytosolic SANT domains of ERj1 that have been shown to associate with the precursors [120]. In addition, ERj1 may have regulatory roles in transcription and/or translation [98,99,100,121]. The cytosolic domain of ERj1 has the ability to allosterically inhibit translation at the stage of initiation when its ER lumenal J-domain is not bound to BiP [100]. Thus, ERj1 would be ideally suited to allow initiation of synthesis of precursor polypeptides on ER-bound ribosomes when BiP is available on the ER lumenal side of the membrane. Furthermore, ERj1 has all the features of a membrane-tethered transcription factor that can be activated by regulated intra-membrane proteolysis, i.e., it has the capacity to bind to DNA as well as RNA [121]. 

After ERj1 depletion with two different targeting siRNAs, 4,947 different proteins were quantitatively characterized by MS, which were detected in all samples (Figure 7, Table 2, Appendix A). The MS data have been deposited to the ProteomeXchange Consortium via the PRIDE partner repository [113] with the dataset identifier: PXD011990 (http://www.proteomexchange.org, (accessed on 2 May 2021)). Applying the same statistical analysis as before, we found that transient and partial ERj1 depletion significantly affected the steady-state levels of 172 proteins: 92 negatively and 80 positively (q < 0.05) (Figure 7a, Appendix A). As expected, ERj1 itself was negatively affected according to western blot (Figure 7b). However, it was not detected in the proteomic data set (Figure 7a, volcano plot). Of the other negatively affected proteins, GO terms assigned close to ~30% to organelles of the endocytic and exocytic pathways (Figure 7c, large pies). We also detected only little enrichment of proteins with SP, N-glycosylated proteins, and membrane proteins (Figure 7c, small pies). Furthermore, the identified precursors included only seven proteins with cleavable SP (i.e., secretory protein complement factor C3, lysosomal proteins CTSD and NEU1, ER proteins FKBP7, SUMF2, and the membrane proteins CD47 and CD58) and eight membrane proteins with TMH (i.e., ATP6V0C, ER proteins BCL2L1, PTPLB, SOAT1, Golgi proteins GALNT4, QPCTL, and plasma membrane proteins COMT, TMEM131) (Appendix A). Of these 15 negatively affected proteins ten were N-glycosylated proteins. Notably, there was no known interaction partner of ERj1 detected among the negatively affected proteins. The positively affected proteins included the cytosolic ubiquitin-conjugating enzymes ISG15 and KCMF1, consistent with accumulation of precursor polypeptides in the cytosol after partial ERj1 depletion for 96 h (Appendix A). 

## 3. Discussion

### 3.1. Discussion of the Experimental Approach

The experimental approach was designed to identify substrates of components, which are involved in protein import into the human ER under physiological conditions, i.e., as compared to more or less artificial situations of in vitro experiments where single precursor proteins are studied one at a time in cell-free systems for translation and protein import into microsomes or the ER of semi-permeabilized cells or under cellular conditions where single precursor proteins are heavily over-produced. This approach represents a combination of siRNA-mediated gene silencing for the protein transport component of interest in human cells, label-free quantitative MS analysis of the total cellular proteome, and subsequent differential protein abundance analysis for two cell pools, which had been treated with two different siRNAs targeting the same gene, compared to a pool of cells, which had been treated with a non-targeting (control) siRNA. Initially, the approach was established for the essential transport component Sec61 complex, which served as a proof-of-principle since it is necessary for the ER import of most precursor proteins (Figure 2 and Figure 3) [101]. First, the timing of the experiment had to be optimized, which began on day one (time 0) with seeding of the cells, was followed by siRNA transfections on days two (24 h) and three (48 h) and was terminated by harvesting of the cells on day four. Except for BiP where the experiment had to be terminated after 72 h, the cells tolerated depletion of the transport component for these 96 h without dramatic changes in cell growth and cell viability. In the case of Sec61 depletion, gene ontology (GO) terms assigned over 60% of the 482 negatively affected proteins to organelles of the endocytic and exocytic pathways, representing a strong enrichment compared to the value for the total quantified proteome (Figure 3b). In addition, significant enrichment of precursor proteins with SP, N-glycosylated proteins, and membrane proteins was detected, and the analysis included proteins with low and high cellular concentrations, ranging from below 1 to almost 10,000 nM (Appendix A). This suggested that the precursors of these proteins are substrates of the Sec61 complex. According to bioinformatic analysis ~30% of the total quantified proteome of roughly 7200 proteins comprises Sec61 substrates. Thus, the experimental approach underestimates the number of different precursor polypeptides that rely on the Sec61 complex. As expected, those numbers were lower for all other transport components since the latter are known to be precursor-specific, i.e., involved in import of only subsets of precursor polypeptides [101,102,103]. In the case of SP-dependent ER protein import, the analysis of SPs of precursors, which were negatively affected by depletion of a certain transport component, allowed to deduce the rules of engagement of the respective components and to conclude that not all SPs are equal and, therefore, have their specific requirements (Figure 8, Figure 9 and Figure 10, Appendix A). We suggested that these features allow for differential regulation of ER import under different cellular conditions, for example, by phosphorylation and de-phosphorylation of transport components, which is known to occur but, so far, has not been analyzed in any detail (Table 1).

The question is how the approach can be improved in future experiments. In the cases of Sec61 and TRAP depletion, the experiments with the nine cell pools were repeated two or three times, which increased the coverage of the cellular proteomes (Table 2) [101]. In the cases of Sec62 and Sec63 depletion, the creation of CRISPR HEK293 cell lines improved the number of negatively affected precursor polypeptides with SP or TMH significantly (i.e., from 18 to 62 and 6 to 22 for Sec62, from 3 to 21 and 6 to 29 for Sec63) (Appendix A) but also gave more time for adaptations [103]. Therefore, additional experiments for the mRNA receptors will have to be carried out and simultaneous knock-down of two components with similar function and potential overlapping substrate specificities will be employed in the strategy [51], as we are currently doing for the SP- and TMH-dependent targeting pathways (Figure 1, upper part). Furthermore, in addition to analysis of the total cellular proteome, the secretome of the different cells shall be analyzed in future experiments, which should focus the strategy on secretory proteins, i.e., precursors with SP and N-glycosylation [122,123]. Notably, for related questions on protein import into mitochondria, the analysis was focused on accumulating precursors in the cytosol, which—in principle—could also be done for protein import into the ER under conditions of proteasome inhibition [124].

### 3.2. Discussion of the Results on Possible Clients for mRNA Targeting to the ER

As expected, the depletion of the essential Sec61 complex had the most pronounced effect on ER import of precursor polypeptides with N-terminal SP (197) or TMH (98) [101]. These numbers dropped to 38 and 22 for TRAP depletion [101], 13 and 17 for TRAM depletion [102], 18 and 6 for Sec62 depletion and 3 and 6 for Sec63 depletion (Table 2 and Table 3) [103]. In the case of BiP depletion the numbers were 33 and 22, respectively, but are overshadowed by the fact that in this case not only ER protein import but protein folding and assembly in the ER were disturbed, which can be deduced from the induction of both an ERAD component (SEL1) as well as a UPR component (ATF6) and UPR-controlled ER chaperones, i.e., in addition to ER protein import components and cytosolic ubiquitin ligases (Appendix A, see Section 3.3) [103]. For the three proteins, which were the focus here, RRBP1 had the most pronounced effect on ER import of precursor polypeptides with N-terminal SP (21) or TMH (18), which is comparable to TRAM and TRAP depletion. In the case of ERj1, the numbers were much lower (7 and 8), i.e., similar to the depletion of its putative paralog Sec62/Sec63. In the case of KTN1, the numbers were similarly low (3 and 8), but, strikingly, cytosolic proteins were over-represented in the negatively affected proteins (44% versus 20% on average; in toto 20), in particular cytoskeletal proteins and protein kinases. Remarkably, there was quite a bit of overlap in potential substrates between the three potential mRNA receptors, in particular between ERj1 and KTN1. The overlap was one protein between RRBP1 and KTN1 (extracellular matrix protein COL4A2), one protein between RRBP1 and ERj1 (membrane protein SOAT1), and six membrane proteins between ERj1 and KTN1 (ATP6V0C, BCL2L1, CD47, GALNT4, PTPLB, QPCTL) (Figure 11). The latter is particularly striking in light of the low numbers of negatively affected proteins of the secretory pathway for these two components. Interestingly, the putative RRBP1 clients DEGS1 and PARL plus the shared KTN1 and ERj1 client BCL2L1 are also found in one of the mitochondrial membranes, which may suggest that some of these two proteins, after targeting of their mRNAs to the ER and their subsequent translation at this location, may travel from the ER to mitochondria via the ER-SURF pathway, which was recently identified in yeast [66].

Taken together, these results are consistent with the view that RRBP1 is an ER protein import component, serving either as an mRNA receptor or as a RNC receptor for SP- and TMH-comprising precursors of soluble as well as membrane proteins. This interpretation is in line with the observations that RRBP1 depletion stimulates SRPRA and SRPRB up-regulation as well as the up-regulation of cytosolic chaperones and ubiquitin ligases. In addition, a function in ER protein import of ERj1 is supported by the proteomic approach but ERj1 appears to be acting in cotranslational protein import rather than in targeting of mRNAs or RNCs. In contrast, it appears that KTN1 may only play a minor or more specialized role in ER protein import. Consistent with this interpretation, all the putative compensatory mechanisms after RRBP1 depletion were not observed after KTN1 depletion. Interestingly, however, RRBP1 was up-regulated under these conditions, suggesting that the two receptors may have some overlap in their client mRNAs, such as in the case of the observed mRNA coding for collagen 4A2. Notably, RRBP1 up-regulation had previously been observed after BiP and TRAM1 depletion (see Section 3.3). Furthermore, the remarkable degree of overlap of 55 and 40%, respectively, in membrane protein substrates with SP (CD47) or TMH (ATP6V0C, BCL2L1, GALNT4, PTPLB, QPCTL) between ERj1 and KTN1 clearly warrants future research into the possible implications. The first and more likely one could be that the two proteins cooperate in the ER import of their common membrane protein clients. Second, the two proteins may just serve as alternative mRNA or RNC receptors for these particular membrane protein clients. The observation that KTN1 may play a more pronounced role for cytosolic proteins, predominantly cytoskeletal proteins and protein kinases, also warrants some discussion and future work. Here, it remains open if the KTN1 interaction with the cytoskeleton may be related to the observed degradations.

When the present results on the two putative mRNA receptors RRBP1 and KTN1 are compared with the putative clients of the putative mRNA receptor LRRC59, only the cytoskeletal protein vinculin was affected by all three mRNA receptors, again pointing to a certain functional overlap (Figure 11) [96]. Among the twenty most enriched LRRC59 clients, which were identified by the combination of ribosome proximity labeling and transcriptome analysis, there are eleven cytosolic proteins (AIMP1, BAG6, COLCA2, DAB1, ISL1, MAP7D2, MARS, PGR, RAB32, RAB3C, **VCL**), three membrane proteins with SP (DNER, LSAMP, PTPRO), and six membrane proteins with TMH (AGTR1, CYP4F22, NRSN1, OPRK1, SLC4A10, SLC24A3) consistent with the proposed role of these three receptor proteins in the biogenesis of cytosolic proteins as well as membrane proteins of the secretory pathway. When the present results are compared with the putative clients of AEG-1 that were characterized by mRNA crosslinking and subsequent transcriptome analysis [90], the common denominators are again membrane proteins of the secretory pathway (including the KTN1 clients ATP6V0C, CD47, GANAB, and TMEM106B plus the RRBP1 clients CPD, CYP51A1, DAG1, DEGS1, DPY19L1, GGH, L1CAM, LAMB1, NCEH1, PARL, PDIA6, SDC1, SLC3A2, SOAT1 SPTLC1, SUN2, TGFBI, and TOR1B) as well as cytosolic proteins (such as the KTN1 client JUP). However, most AEG-1 clients were found to be mRNAs coding for organelle resident proteins and the interaction sites in the mRNAs were found to be in the coding regions rather than anyone of the two untranslated regions (UTRs), i.e., AEG-1 appears to be an RNC rather than an mRNA receptor.

To address the substrate spectrum of the three proteins of this study in more detail, we first analyzed the data for precursor polypeptides, which were negatively affected by depletion of the two dedicated ER protein import components RRBP1 and ERj1 with respect to the physico-chemical properties of their SPs and TMHs. The rationale was that theses SPs and TMHs may have features that interfere with efficient SRP dependent targeting. Using established custom scripts [101,102,103], we computed the hydrophobicity score of SP of RRBP1- (*n* = 21) and ERj1 clients (*n* = 7), glycine/proline (GP) content, and delta G_app_ values of the same SP as described in Materials and Methods (Section 4.4. Data analysis). All values were plotted against the relative count (Figure 12a,c). Additional plots were computed for TMHs (*n* = 17 and 8, respectively) (Figure 12b,d). We also used custom scripts to extract all SP and TMH annotations for human proteins from UniProtKB entries and subjected them to the same calculations. None of the analyses led to any significant differences between the clients and the total human SPs and TMHs (only delta G values are shown), which would have suggested a reason for the observed dependencies of the putative clients.

Therefore, the mRNAs of the three novel substrate data sets were analyzed next. Here, the focus was on AU-rich elements (AREs), specifically ATTTA motifs, in the 3′UTRs, which are known to play a role for example in mRNA stability. Although these motifs are present in many mRNAs, they were of interest here since multiple ATTTA motifs in the 3′UTRs in combination with coding regions for transmembrane domains had been shown to be involved in the biogenesis of the KTN1- and ERj1-client CD47 as well as two additional membrane proteins, BCL2 and PD-L1 (also termed CD274) [125,126]. First, we asked how many of the RRBP1, KTN1, and ERj1 clients have ATTTAs in their mRNAs (Appendix A). AREs in 3′UTRs were identified in 200 out of 298 genes corresponding to proteins affected in the RRBP1 depletion experiment (67.1%). Similarly, 67.1% (47 out of 70) and 63.9% (110 out of 172) of genes corresponding to proteins affected in the KTN1 and ERj1 depletion experiments were found to be ARE-containing genes, respectively. For comparison, the percentages values are 69.2% (243 out of 351), 76.5% (26 out of 34), 71.6% (184 out of 257) in the case Sec62, Sec63, and TRAP, respectively. Notably, the percentage of ARE-containing genes for SP- and TMH-dependent targeting (affected by Sec61, Sec63, and TRAP depletion) is significantly ~2% higher than SP- and TMH-independent targeting. Thus, the global analysis did not lead to any insights for the problem at hand. Next, we asked how many of the RRBP1, KTN1, and ERj1 clients have multiple ATTTAs in their mRNAs and code for membrane proteins of the secretory pathway (Appendix A). After RRBP1 depletion, 23 membrane proteins were negatively affected (not counting RRBP1), 15 contained AREs, one contained more than fifteen AREs (ATL3 with 18, CPD with 13, and SOAT1 with 11), 10 contained five or more AREs (including ATL2 with 5) (Appendix A). Thus, 65% of the negatively affected membrane proteins contained ATTTA motifs and 7% of these comprised more than fifteen ATTTAs, which we defined here as multiple AREs. Following KTN1 depletion, eight membrane proteins were negatively affected (not counting KTN1), five contained AREs, three of these contained more than fifteen AREs (CD47 with 16, GALNT4 with 19, and TMEM106B with 43). Thus, 63% of the negatively affected membrane proteins contained ATTTA motifs and 60% of these comprised multiple AREs. After ERj1 depletion, 10 membrane proteins were negatively affected, eight contained AREs, two contained more than fifteen AREs (CD47 and GALNT4). Thus, 80% of the negatively affected membrane proteins contained ATTTA motifs and 25% of these comprised more than fifteen ATTTA motifs. Although the numbers are low, there appears to be a trend that KTN1 membrane protein clients may have multiple ATTTA motifs in 3′UTRs of their mRNAs, i.e., KTN1 may be an mRNA receptor for ARE-containing mRNAs which will be discussed below in 3.4. Notably, the mRNA coding for KTN1 contains only three ATTTA motifs in its 3′UTR, i.e., it does not contain multiple AREs (Table 1).

### 3.3. Discussion of Compensatory Mechanisms after Depletion of mRNA Receptors on the ER

In addition to negatively affected proteins, the analysis of the total cellular proteomes in response to depletion of certain ER protein import components also gives important clues, which warrant to be addressed by future work. In general, the severity of the negative impact on precursor proteins correlated with the strength of the positive impact on cytosolic components, which are involved in stabilizing proteins prone to mis-folding and aggregation, the cytosolic chaperones (most pronounced in the case of Sec61 depletion), and on cytosolic components, which are involved in proteasomal degradation, the ubiquitin ligases (in the case of all depletions except for KTN1 and Sec63) (Appendix A) [101]. Similarly, the observed up-regulation of components for protein import into mitochondria, such as TOMM6 and TOMM7 plus various TIMMs, by depletion of Sec61 and BiP points to an additional way for the cell to avoid proteostatic trouble in the cytosol, which is in perfect line with our previous observations that some ER-targeted precursor polypeptides enter mitochondria in the absence of proper ER targeting [127]. But there were also differences between BiP and Sec61 depletion: in the first case ER chaperones were up-regulated by the unfolded protein response (UPR), while in the second case cytosolic chaperones were up-regulated (see above), highlighting that in the first case ER protein import plus protein folding and assembly within the ER were negatively affected, while in the second it was only protein import (Appendix A) [101,103]. The question is if there are dedicated signal transduction pathways in addition to UPR for these apparent regulatory phenomena. Alternatively, the up-regulation may not be due to stimulated transcription. In fact, in the case of SRA and SRB up-regulation after TRAP depletion there was no increase of the corresponding mRNAs detected by quantitative RT-PCR [101]. Therefore, it was suggested that the up-regulation is due to either increased protein synthesis or stability. This requires further qRT-PCR analyses.

Furthermore, exciting possible genetic interactions or other compensatory mechanisms between different pathways for ER targeting and ER insertion or translocation of precursor proteins became visible: under conditions of depletion of Sec61, RRBP1, BiP, Sec62, TRAM1, and TRAP the protein targeting components SRPRA and SRPRB were up-regulated, plus in the case of Sec61- and BiP-depletion several SRP subunits, plus in the case of Sec61-depletion a subunit of the Bag6 complex (TRC35), which is all in line with our previous observations that protein targeting pathways to the ER have overlapping specificities (Appendix A) [49,50,51,52]. This was extended here to mRNA targeting by RRBP1 and, possibly, to the membrane protein insertase termed EMC. Furthermore, RRBP1 was found to be up-regulated after KTN1 depletion, as had previously been observed after depletion of BiP and TRAM1 (Appendix A). Thus, there may exist some kind of genetic mechanism that senses problems in the early steps of the secretory pathway—i.e., ER targeting and ER membrane insertion and translocation—and increases the capacity of the involved mechanisms. That should also be an interesting path to follow in future research.

### 3.4. Possible Implications for the TIGER Domain

The RNA-binding protein TIS11B was described to form a cytosolic micro-domain, which was called TIS granule that enriches membrane protein-encoding mRNAs with multiple AREs in their 3′UTRs in the neighborhood of the ER and, therefore, was called TIGER domain (Figure 1) [125,126,128,129]. This cytosolic sub-domain in the vicinity of the ER enables formation of specific and functionally relevant protein-protein interactions that cannot be established outside. This was first demonstrated for the plasma membrane protein CD47 [127]. Briefly, CD47 is encoded by either a long or a short mRNA, which are distinguished by different 3′UTRs. Only the mRNA with the long and ARE-rich 3′UTR interacts with the cytosolic RNA-binding protein TIS11B, which directs this mRNA to the TIGER domain where the cytosolic domains of newly synthesized and membrane-integrated CD47 interacts with the highly acidic cytosolic protein SET. The latter interaction allows for more efficient CD47 plasma membrane expression. The details of how the TIGER domain creates its special environment with distinct biochemical and biophysical properties is unknown as is the putative mRNA receptor in the ER membrane.

On the basis of the data, which were presented here, the following scenario seems plausible. KTN1 may be the ER-membrane-resident mRNA-binding protein that is enriched in the TIGER domain, where it takes over mRNAs from TIS11B and allows initiation of their translation by Sec61-bound ribosomes. If the mRNA codes for the precursor of a membrane protein with a N-terminal SP (such as CD47) or a membrane protein with an N-terminal TMH (such as GALNT4 and TMEM106B) the nascent precursor begins to sample the cytosolic funnel of the Sec61 channel, which leads to spontaneous channel opening or the recruitment of auxiliary factors such as the TRAP or Sec62/Sec63 complex. Since ERj1 was found to have overlapping substrate specificities with KTN1, it may co-operate with KTN1 in allowing Sec61 channel-opening when BiP is bound to its J-domain [97,98,99,100]. Next, the precursor is imported into the ER or integrated into the ER membrane (CD47). In the case of CD47 the cytosolic protein SET would bind and facilitate plasma membrane expression. If the mRNA codes for a cytosolic protein (such as the cytoskeletal protein vinculin or the protein kinase PDPK1), sampling of the Sec61 channel remains unproductive and the heterodimeric cytosolic protein NAC gets access to the N-terminus of the nascent polypeptide and leads to its release from Sec61 and the simultaneous release of the ribosome from Sec61 [130]. Next, translation of the cytosolic protein would be completed and the protein would be enriched in the TIGER domain where it may play its physiological role. Interestingly, vinculin and PDK1 play a role in focal adhesion. Therefore, it is tempting to speculate that intracellular targeting of cytosolic proteins such as vinculin and PDK1 may also be aided by their synthesis in TIGER domains and interaction with distinct binding partners, in analogy to surface expression of CD47. This clearly warrants to be tested in future work.

## 4. Materials and Methods

### 4.1. Materials

SuperSignal West Pico Chemiluminescence Susbtrate (# 34078) was purchased from Pierce^TM^, Thermo Fisher Scientific, Darmstadt, Germany. ECL^TM^ Plex goat anti-rabbit IgG-Cy5 (PA45011, used dilution 1:1000), and ECL^TM^ Plex goat anti-mouse IgG-Cy3 conjugate (PA43009, used dilution 1:2500) were purchased from GE Healthcare, Freiburg, Germany. Horseradish peroxidase coupled anti-rabbit IgG from goat (A 8275, used dilution 1:1000) was from Sigma-Aldrich, Taufkirchen, Germany. We also purchased from Sigma-Aldrich murine monoclonal antibodies against β-actin (A5441, used dilution 1:10,000) and rabbit polyclonal antibodies against KTN1 or RBBP1 (Sigma-Aldrich, HPA003178 or HPA011924, used dilutions 1:500). Murine monoclonal antibody against SRP54 was purchased from BD Biosciences, Heidelberg, Germany (610940, used dilution 1:1000). Antibodies against ERj1, Sec62, and SRα were raised against the C-terminal oligopeptide of ERj1 or Sec62 plus an additional N-terminal cysteine (CELVQKKKQAKS or CGETPKSSHEKS in single letter code, used dilution 1:500) and an internal peptide of SRα plus a C-terminal cysteine (KKFEDSEKAKKPVRC, used dilution 1:500), respectively, as previously described.

### 4.2. Cell Manipulation and Analysis

HeLa cells (DSM no. ACC 57) were obtained from the German Collection of Microorganisms and Cell Cultures GmbH, Braunschweig, Germany, routinely tested for mycoplasma contamination by VenorGeM Mycoplasm Detection Kit (Biochrom AG, WVGM, Berlin, Germany), and replaced every five years by a new batch. They were cultivated at 37 °C in a humidified environment with 5% CO_2_, in DMEM with 10% fetal bovine serum (FBS; Sigma-Aldrich) and 1% penicillin and streptomycin. The cells are routinely kept in culture for up to thirty passages before a new vial is thawed and they are employed in experiments after five passages. Cell growth was monitored using the Countess^®^ Automated Cell Counter (Invitrogen, Thermo Fisher Scientific) following the manufacturer’s instructions.

For gene silencing, 5.2 × 10^5^ HeLa cells were seeded per 6-cm culture plate, followed by incubation under normal culture conditions. Next, the cells were transfected with a final concentration of 25 nM targeting siRNA (Qiagen, Hilden, Germany) or 25 nM AllStars Negative Control siRNA (Qiagen) using HiPerFect Reagent (Qiagen) following the manufacturer’s instructions. After 24 h, the medium was changed and the cells were transfected a second time. Thus, in each case, silencing was performed for a total of 96 h using two different siRNAs. The targeting siRNAs had the following sequences:
*RRBP1*siRNA#1, GGAUAUUUACGACACUCAAdTdT;*RRBP1*siRNA#2, GAGAUUGUAGAGAAGCUAAdTdT;*KTN1*siRNA#3, CAGUUGGAGCAAAGACUAAdTdT;*KTN1*iRNA#4, GCCUCUGACUUCAACUCAAdTdT;*ERJ1*siRNA#5, CCUCAAUAUUUCUACGUCAdTdT;*ERj1*siRNA#6, GGUAUGAUGAUAUUCUGAUdTdT.

Silencing efficiencies were evaluated by western blot analysis using the appropriate antibodies and an anti-β-actin antibody from mouse. Primary antibodies were visualized with ECL^TM^ Plex goat anti-rabbit IgG-Cy5 (ERj1, SRα) or ECL^TM^ Plex goat anti-mouse IgG-Cy3 conjugate (ß-actin, SRP54) using the Typhoon-Trio imaging system combined with Image Quant TL software 7.0 (GE Healthcare, Freiburg, Germany). Alternatively, peroxidase-coupled anti-rabbit IgG was employed in combination with SuperSignal West Pico Chemiluminescent Substrate and the Fusion SL (peqlab, Erlangen, Germany) luminescence imaging system with accompanying software (KTN1, RRBP1, Sec62).

### 4.3. Label-Free Quantitative Proteomic Analysis

After growth for 96 h, 1 × 10^6^ cells (corresponding to roughly 0.2 mg protein) were harvested, washed twice in PBS, and lysed in buffer containing 6 M GnHCl, 20 mM tris(2-carboxyethyl)phosphine (TCEP; Pierce^TM^, Thermo Fisher Scientific), 40 mM 2-chloroacetamide (CAA; Sigma-Aldrich) in 100 mM Tris, pH 8.0 [101]. The lysate was heated to 95 °C for 2 min, and then sonicated in a Bioruptor sonicator (Diagenode, Seraing, Belgium) at the maximum power setting for 10 cycles of 30 s each. For a 10% aliquot of the sample, the entire process of heating and sonication was repeated once, and then the sample was diluted 10-fold with digestion buffer (25 mM Tris, pH 8, 10% acetonitrile). Protein extracts were digested for 4 h with Lysyl endoproteinase Lys-C (Wako Bioproducts, Fujifilm, Neuss, Germany, enzyme to protein ratio: 1:50), followed by the addition of trypsin (Promega, Heidelberg, Germany) for overnight digestion (enzyme to protein ratio: 1:100). The next day, booster digestion was performed for 4 h using an additional dose of trypsin (enzyme to protein ratio: 1:100). After digestion, a 10% aliquot of peptides (corresponding to about 2 µg of peptides) were purified via SDB-RPS StageTips [104], eluted as one fraction, and loaded for mass spectrometry analysis. Purified samples were loaded onto a 50 cm column (inner diameter: 75 microns; packed in-house with ReproSil-Pur C18-AQ 1.9-micron beads, Dr. Maisch HPLC GmbH, Ammerbuch, Germany) via the autosampler of the Thermo Easy-nLC 1000 (Thermo Fisher Scientific) at 60 °C. Using the nanoelectrospray interface, eluting peptides were directly sprayed onto the benchtop Orbitrap mass spectrometer Q Exactive HF (Thermo Fisher Scientific) [105]. Peptides were loaded in buffer A (0.1% (*v*/*v*) formic acid) at 250 nL/min and the percentage of buffer B was ramped to 30% over 180 min, followed by a ramp to 60% over 20 min, then 95% over the next 10 min, and maintained at 95% for another 5 min [103]. The mass spectrometer was operated in a data-dependent mode with survey scans from 300 to 1700 *m*/*z* (resolution of 60,000 at *m*/*z* = 200). Up to 15 of the top precursors were selected and fragmented using higher energy collisional dissociation (HCD) with a normalized collision energy value of 28 [103]. The MS2 spectra were recorded at a resolution of 17,500 (at *m*/*z* = 200). AGC target for MS and MS2 scans were set to 3E6 and 1E5, respectively, within a maximum injection time of 100 and 25 ms for MS and MS2 scans, respectively. Dynamic exclusion was enabled to minimize repeated sequencing of the same precursor ions and set to 30 s [103].

### 4.4. Data Analysis

Each MS experiment provided proteome-wide abundance data as LFQ intensities for three sample groups—one control (non-targeting siRNA treated) and two stimuli (down-regulation by two different targeting siRNAs directed against the same gene)—each having three data points. Missing data points were generated by imputation, whereby we distinguished two cases. For completely missing proteins lacking any valid data points, imputed data points were randomly generated in the bottom tail of the whole proteomics distribution, following the strategy in the Perseus software (http://maxquant.net/perseus/ last accessed on 2 May 2021) [109]. For proteins having at least one valid MS data point, missing data points were generated from the valid data points based on the local least squares (LLS) imputation method [110]. The validity of this approach is demonstrated [101]. Subsequent to data imputation, gene-based quantile normalization was applied to homogenize the abundance distributions of each protein with respect to statistical properties.

Protein annotations of SP, transmembrane regions, and N-glycosylation sites in humans were extracted from UniProtKB entries using custom scripts. Using custom scripts, we computed the hydrophobicity score and glycine/proline (GP) content of SP and TMH sequences. A peptide’s hydrophobicity score was assigned as the average hydrophobicity of its amino acids according to the Kyte-Doolittle propensity scale (averaged over the sequence length) [101,103]. GP content was calculated as the total fraction of glycine and proline in the respective sequence [101]. G_app_ values of SP and TMH were calculated with the G_app_ predictor for TM helix insertion (https://dgpred.cbr.su.se/index.php?p=home last accessed on 2 May 2021). We also used custom scripts to extract all SP annotations for human proteins from UniProtKB entries (human) and subjected them to the same calculations.

The AREsite2 database (http://rna.tbi.univie.ac.at/AREsite2/welcome last accessed on 2 May 2021) and an in-house script were used to identify AU-rich elements (AREs) in human genes [129]. Ensemble gene IDs corresponding to gene symbols were obtained from the bioDBnet database (http://biodbnet.abcc.ncifcrf.gov last accessed on 7 May 2021) [131].

## 5. Conclusions

The novel experimental approach, which was described here, was designed to identify substrates of components that are involved in protein import into the human ER under physiological cellular conditions. It represents a combination of siRNA-mediated gene silencing for the protein transport component of interest with two different targeting siRNAs in comparison to a non-targeting siRNA in human cells (in triplicates), label-free quantitative MS analysis of the total cellular proteome, and subsequent differential protein abundance analysis. Originally, it was successfully employed for the essential transport component Sec61 complex, which served as a proof-of-principle, and next applied to additional transport components (Sec62, Sec63, TRAM, TRAP). Here, the approach was used for the functional analysis of putative mRNA or RNC receptors of the ER membrane, RRBP1 plus KTN1, and for ERj1. In the case of RRBP1, a role as mRNA or RNC receptor in the biogenesis of proteins that are destined for the secretory pathway was demonstrated by both the negative effect on various respective precursor proteins as well as the positive effect on SRA and SRB following depletion of RRBP1. In contrast, it turned out that not only proteins that have to be imported into the ER but also cytosolic proteins are degraded in the absence of the mRNA receptor KTN1. Alternatively, the respective mRNAs may have been degraded, for example, by AU-rich element-mediated decay. This implies that KTN1 can also play a role in the biogenesis of proteins that are destined for the secretory pathway, which was demonstrated by the negative effect on certain precursor proteins (such as CD47) plus the positive effect on RRBP1 after KTN1 depletion. Furthermore, the negative effects on CD47 and certain cytoskeletal proteins and protein kinases suggested a possible function of KTN1 as the elusive ER membrane resident mRNA receptor in the so-called TIGER domain. For these cytosolic proteins it may be important to be produced and concentrated near their site of action rather than distributed throughout the cytosol, in particular for membrane-interacting cytoskeletal elements, such as junction plakoglobin and vinculin at adherens junctions between neighboring cells. Interestingly, this may be similarly true for certain protein kinases, such as OXSR1 that plays a role in regulating the actin cytoskeleton in response to environmental stress, PAK1 that regulates cytoskeletal reorganization for cell motility and morphology, and 3-phosphoinositide-dependent protein kinases 1 and 2 (PDPK1 and 2) that are also located at cell junctions. For ERj1, a function in ER protein import is supported by the proteomic approach, but it appears to be acting in co-translational protein import and, eventually, Sec61 channel gating rather than in the targeting of mRNAs or RNCs.

## Figures and Tables

**Figure 1 molecules-26-03591-f001:**
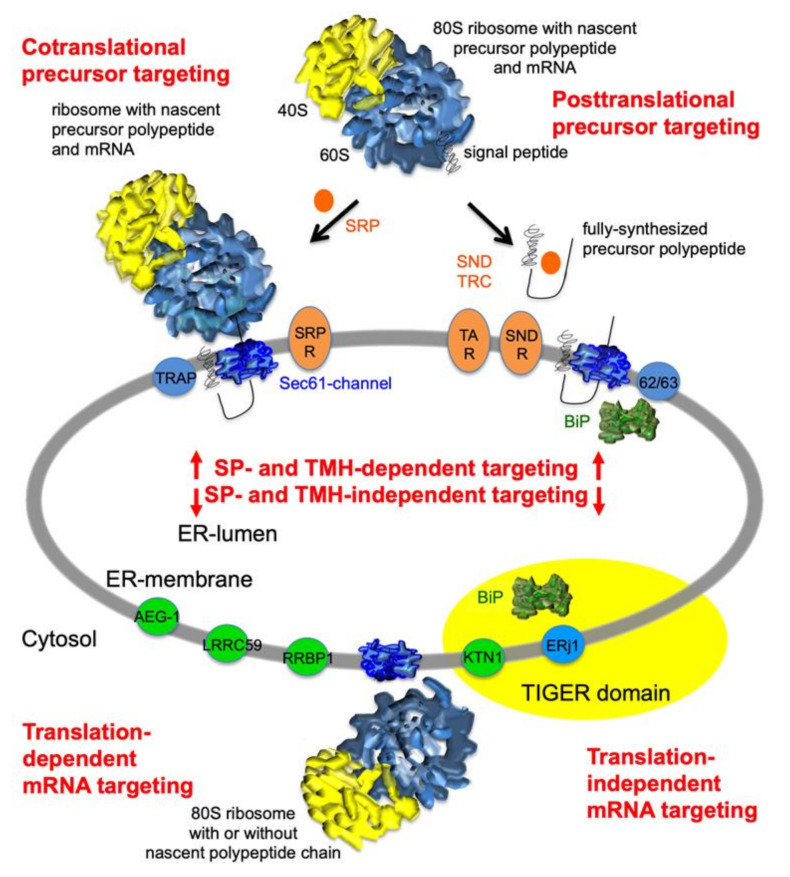
Mechanisms of protein and mRNA targeting to the human endoplasmic reticulum (ER). Import of most precursor polypeptides into the ER involves the heterotrimeric Sec61 channel in the ER membrane, which can facilitate membrane insertion of membrane proteins with N-terminal transmembrane helices (TMHs) and translocation of soluble proteins with N-terminal signal peptides (SPs). Most SPs and TMHs of nascent precursor polypeptides are targeted to the ER membrane by SRP and its heterodimeric receptor in the ER membrane (SRPR), others are targeted post-translationally by the TRC or SND pathway. In addition, there are mechanisms for mRNA targeting to the ER, which either target mRNA to ER-bound ribosomes or mRNA that is present in ribosome-nascent chain (RNC) complexes with chains too short to involve their SPs or TMHs in the targeting reaction. The putative receptors for these mechanisms in the ER membrane are AEG-1, LRRC59, RRBP1, and KTN1 (shown as green circles) and are supposed to also bind unrelated mRNAs. Additional ER membrane proteins support Sec61 channel gating (shown as blue circles).

**Figure 2 molecules-26-03591-f002:**
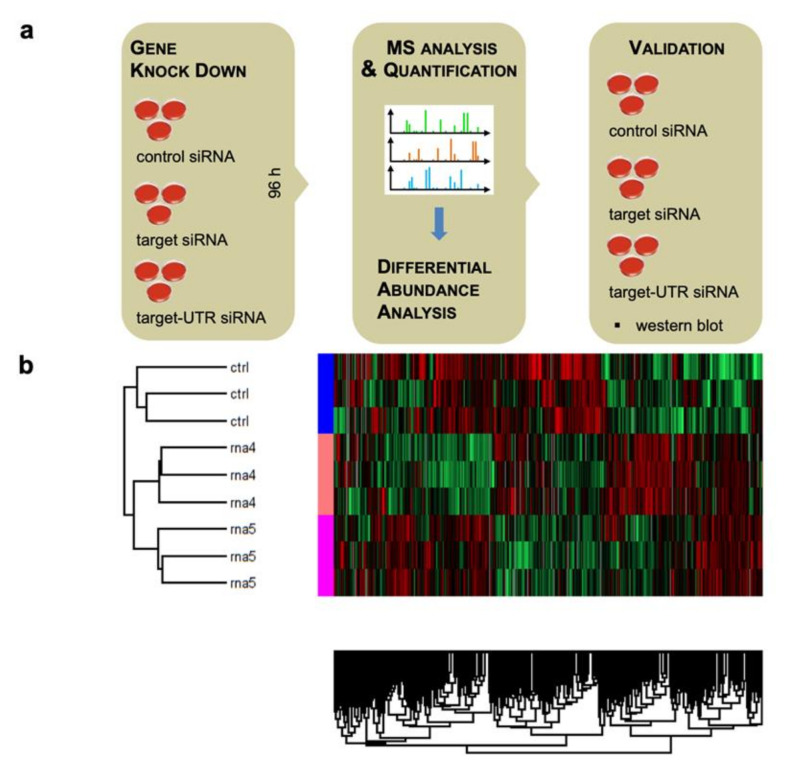
Experimental strategy and representative heat map. (**a**) The experimental strategy was as follows: siRNA-mediated gene silencing using two different siRNAs for each target and one non-targeting (control) siRNA, respectively, with three replicates for each siRNA for 96 h; label-free quantitative analysis of the total cellular proteome; differential protein abundance analysis to identify negatively affected proteins (i.e., putative clients of the target) and positively affected proteins (i.e., putative compensatory mechanisms); validation by western blot. (**b**) Heat maps visualize clusters of proteins that were positively affected following treatment with both siRNAs directed against target mRNA or with non-targeting (control) siRNA, or that were negatively affected following treatment with both siRNAs, or that represent variations between siRNAs. Red indicates positively affected proteins, green indicates negatively affected proteins. The representative heat map shows results from an experiment where *SEC61A1* was the target.

**Figure 3 molecules-26-03591-f003:**
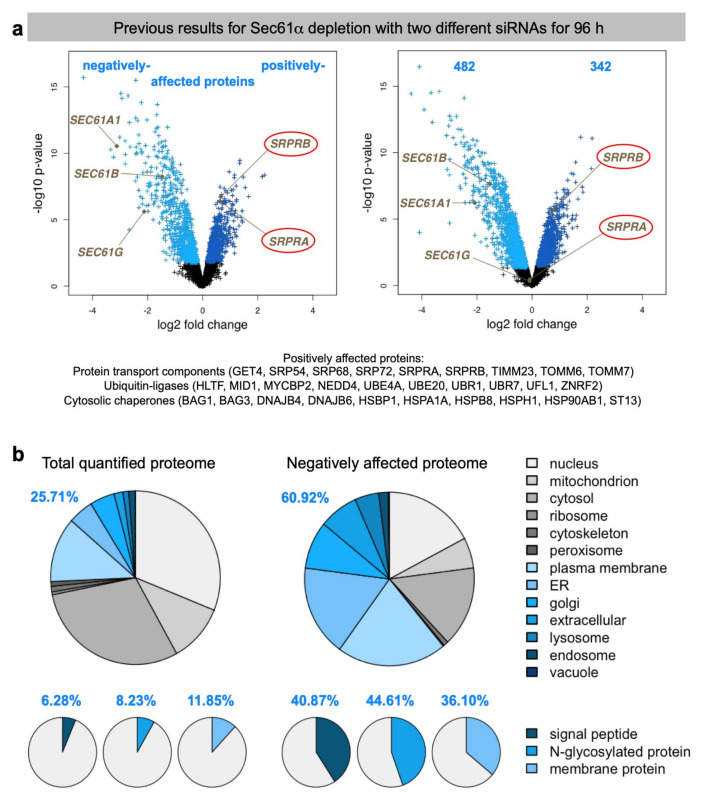
(on previous page)**.** Representative volcano plots and gene ontology (GO) enrichment after Sec61 depletion. (**a**) Differentially affected proteins were characterized by the mean difference of their intensities plotted against the respective permutation false discovery rate-adjusted *p*-values in volcano plots. The results for a single targeting siRNA are shown in each panel. Subunits of the Sec61 complex and of the SRP receptor are highlighted in the plots. In addition, the numbers of proteins, which were negatively or positively affected by both targeting siRNAs, are given in the right panel; the full set of relevant positively affected proteins are given below the plots. (**b**) Classification of Sec61 clients was based on GO enrichment factors where the results from the complete set of quantified proteins in the left panel are compared with the negatively affected proteome. Protein annotations of signal peptides, membrane location, and N-glycosylation in humans were extracted from UniProtKB, and used to determine the enrichment of GO annotations among the negatively affected proteins. The figure summarizes results from an experiment, which served as proof-of-principle for the approach and is shown here in modified form for comparison with the results on putative mRNA receptors [101].

**Figure 4 molecules-26-03591-f004:**
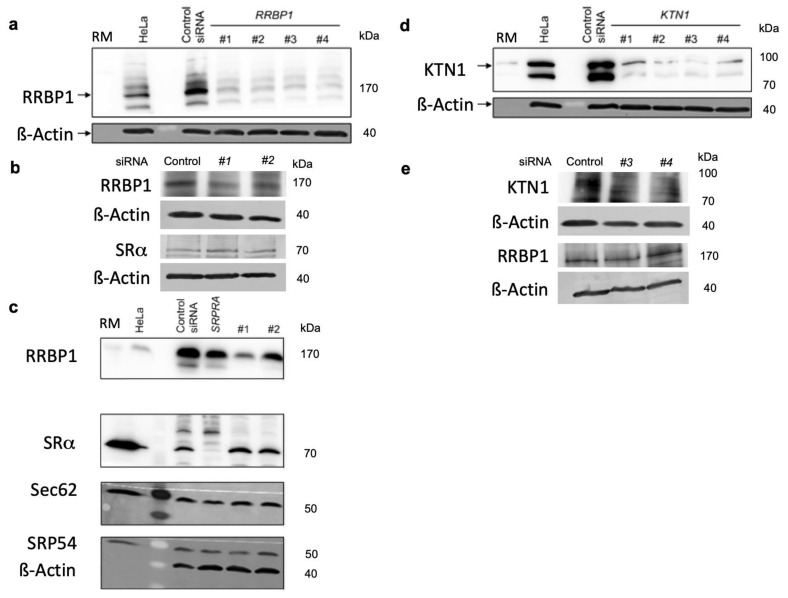
Western blots after RRBP1 or KTN1 depletion. (**a**,**d**) Depletion of RRBP1 or KTN1 was carried out in HeLa cells using four different targeting siRNAs against each target (#1–#4) in comparison to a non-targeting siRNA (control siRNA) for 96 h. Knock-down efficiencies were evaluated by western blot. For the following MS experiments siRNAs #1 and #2 were chosen for depletion of RRBP1 (**a**), siRNAs #3 and #4 for KTN1 depletion (**d**). (**b**,**e**) MS data after depletion of RRBP1 or KTN1 were evaluated with respect to positively affected proteins by western blots. (**c**) Furthermore, MS data after depletion of RRBP1 were evaluated with respect to positively affected and unaffected proteins, respectively, by western blots. (**a**–**e**) Molecular mass values are indicated in kilodaltons (KDa). Only the area of interest of the blots is shown. RM, canine pancreatic rough microsomes, which were used for antigen identification. SRα was depleted from HeLa cells by siRNA treatment as an additional control (SRPRA).

**Figure 5 molecules-26-03591-f005:**
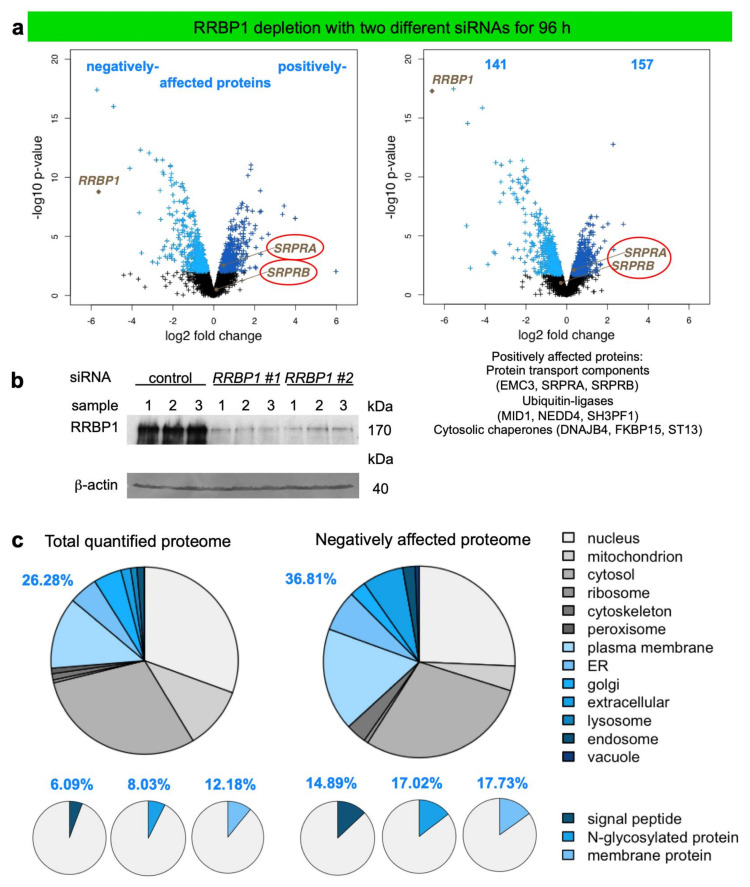
Volcano plots and GO enrichment after RRBP1 depletion. (**a**) Differentially affected proteins were characterized by the mean difference of their intensities plotted against the respective permutation false discovery rate-adjusted *p*-values in volcano plots. The results for a single targeting siRNA are shown in each panel. RRBP1 and the subunits of the SRP receptor are highlighted in the plots. In addition, the numbers of proteins, which were negatively or positively affected by both targeting siRNAs, are given in the right panel; the full set of relevant positively affected proteins are given below the right plot. (**b**) Knock-down efficiencies were evaluated by western blot. Molecular mass values are indicated in kilodaltons (KDa). Only the area of interest of the blot is shown, the original images are shown in Appendix A. (**c**) Classification of RRBP1 clients was based on GO enrichment factors where the results from the complete set of quantified proteins in the left panel are compared with the negatively affected proteome. Protein annotations of signal peptides, membrane location, and N-glycosylation in humans were extracted from UniProtKB, and used to determine the enrichment of GO annotations among the negatively affected proteins.

**Figure 6 molecules-26-03591-f006:**
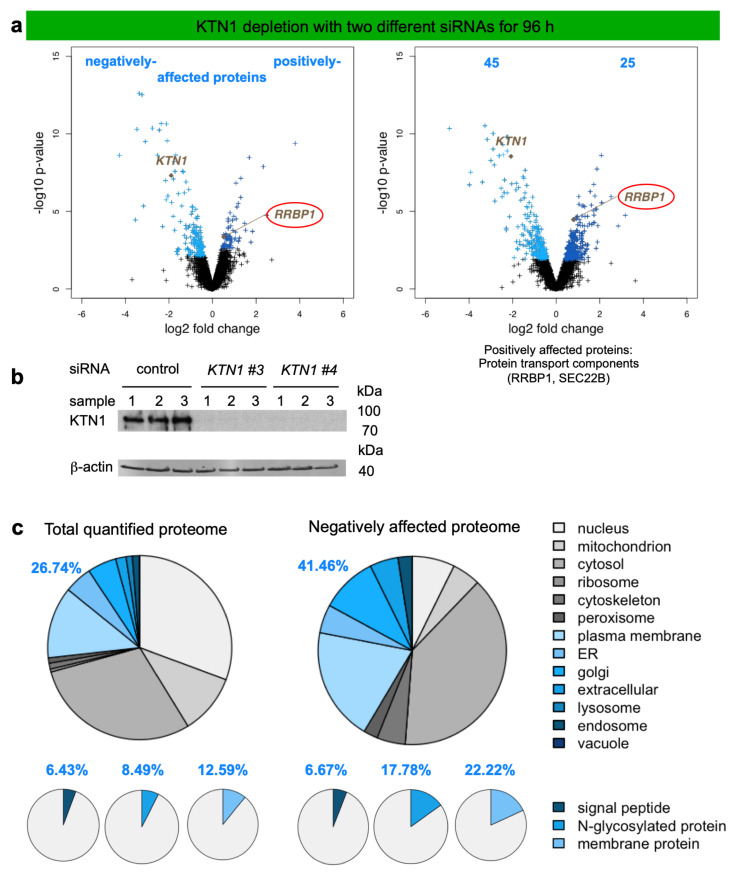
Volcano plots and GO enrichment after KTN1 depletion. (**a**) Differentially affected proteins were characterized by the mean difference of their intensities plotted against the respective permutation false discovery rate-adjusted *p*-values in volcano plots. The results for a single targeting siRNA are shown in each panel. KTN1 and RRBP1 are highlighted in the plots. The numbers of proteins, which were negatively or positively affected by both targeting siRNAs, are given in the right panel; the full set of relevant positively affected proteins are given below the right plot. (**b**) Knock-down efficiencies were evaluated by western blot. Results are presented as described in the legend to Figure 5. (**c**) Classification of KTN1 clients was based on GO enrichment factors as described in the legend to Figure 5.

**Figure 7 molecules-26-03591-f007:**
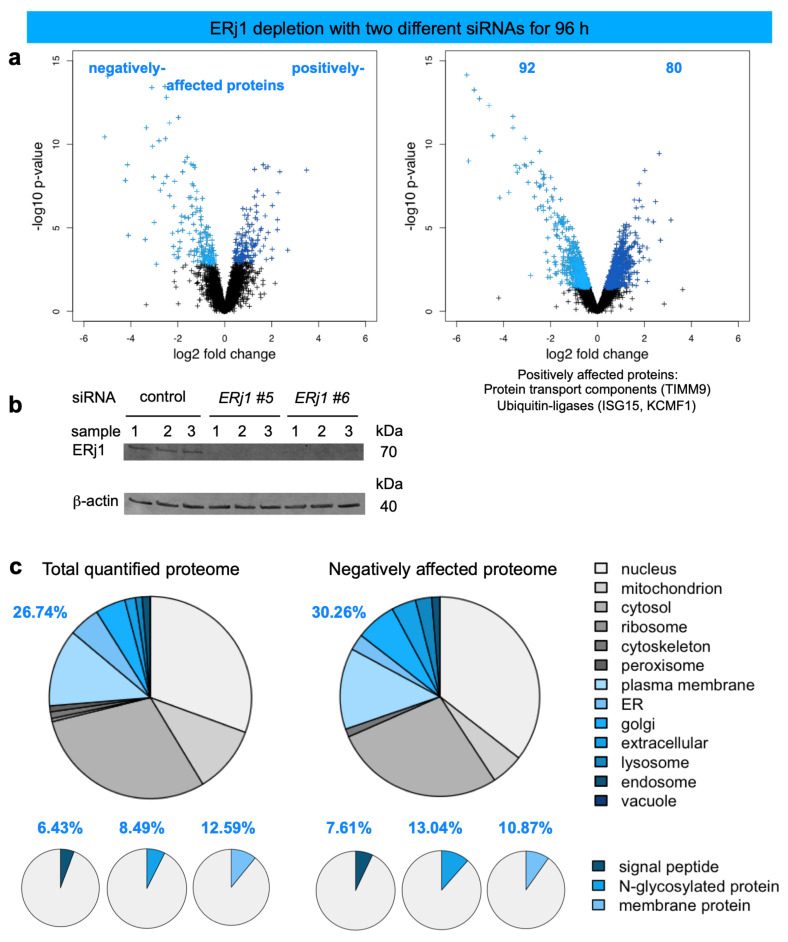
Volcano plots and GO enrichment after ERj1 depletion. (**a**) Differentially affected proteins were characterized by the mean difference of their intensities plotted against the respective permutation false discovery rate-adjusted *p*-values in volcano plots. The results for a single targeting siRNA are shown in each panel. ERj1 was not among the quantified proteins. The numbers of proteins, which were negatively or positively affected by both targeting siRNAs, are given in the right panel; the full set of relevant positively affected proteins are given below the right plot. (**b**) Knock-down efficiencies were evaluated by western blot. Molecular mass values are indicated in kilodaltons (KDa). Only the area of interest of the blot is shown. (**c**) Classification of ERj1 clients was based on GO enrichment factors where the results from the complete set of quantified proteins in the left panel are compared with the negatively affected proteome. Protein annotations of signal peptides, membrane location, and N-glycosylation in humans were extracted from UniProtKB, and used to determine the enrichment of GO annotations among the negatively affected proteins.

**Figure 8 molecules-26-03591-f008:**
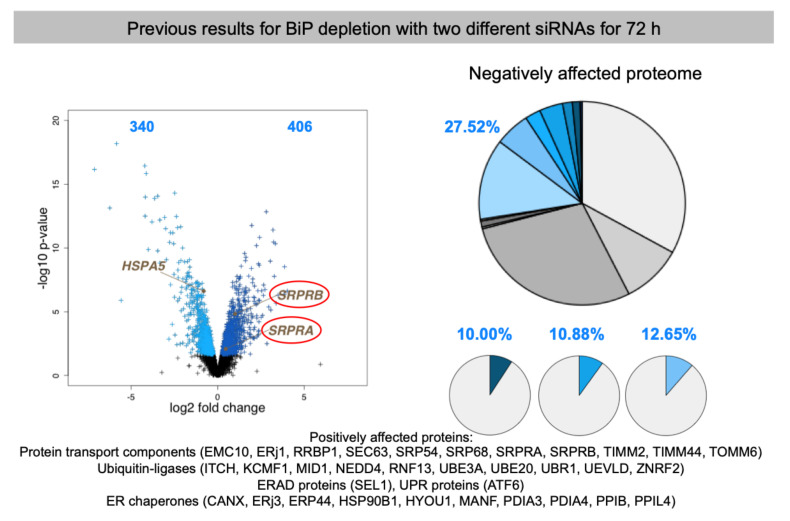
Representative volcano plot and GO enrichment after BiP depletion. Differentially affected proteins were characterized by the mean difference of their intensities plotted against the respective permutation false discovery rate-adjusted *p*-values in volcano plots. The representative results for a single targeting siRNA are shown on the left. BiP, which is coded by the *HSPA5* gene, and subunits of the SRP receptor are highlighted in the plot. In addition, the numbers of proteins, which were negatively or positively affected in both experiments, are given; the full set of relevant positively affected proteins are given below. Classification of BiP clients was based on GO enrichment factors where the results from the complete set of quantified proteins (not shown) are compared with the negatively affected proteome. Protein annotations of signal peptides, membrane location, and N-glycosylation in humans were extracted from UniProtKB, and used to determine the enrichment of GO annotations among the negatively affected proteins. The figure summarizes results from a previous experiment and is shown here in modified form for comparison [103].

**Figure 9 molecules-26-03591-f009:**
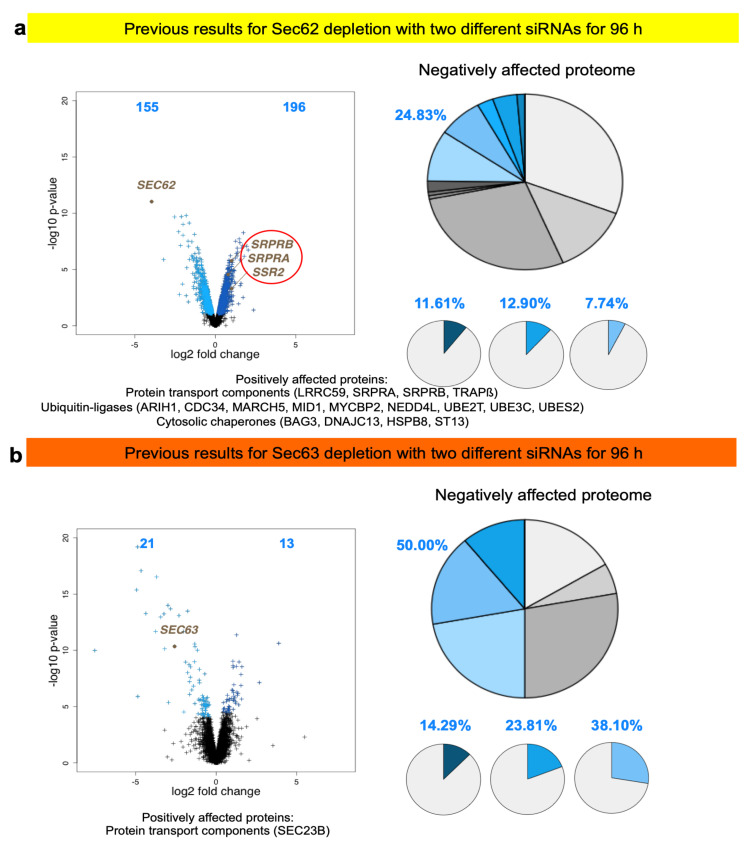
Representative volcano plots and GO enrichment after Sec62 and Sec63 depletion, respectively. Differentially affected proteins were characterized by the mean difference of their intensities plotted against the respective permutation false discovery rate-adjusted *p*-values in volcano plots. The representative results for a single targeting siRNA are shown on the left. *SEC62* (**a**) and *SEC63* (**b**), *SSR2*, which codes for TRAPß, and subunits of the SRP receptor are highlighted in the plot after Sec62 depletion. In addition, the numbers of proteins, which were negatively or positively affected in both experiments, are given; the full set of relevant positively affected proteins are given below. Classification of the respective clients was based on GO enrichment factors where the results from the complete set of quantified proteins (not shown) are compared with the negatively affected proteome. Protein annotations of signal peptides, membrane location, and N-glycosylation in humans were extracted from UniProtKB, and used to determine the enrichment of GO annotations among the negatively affected proteins. The figure summarizes results from previous experiments and is shown here in modified form for comparison [103].

**Figure 10 molecules-26-03591-f010:**
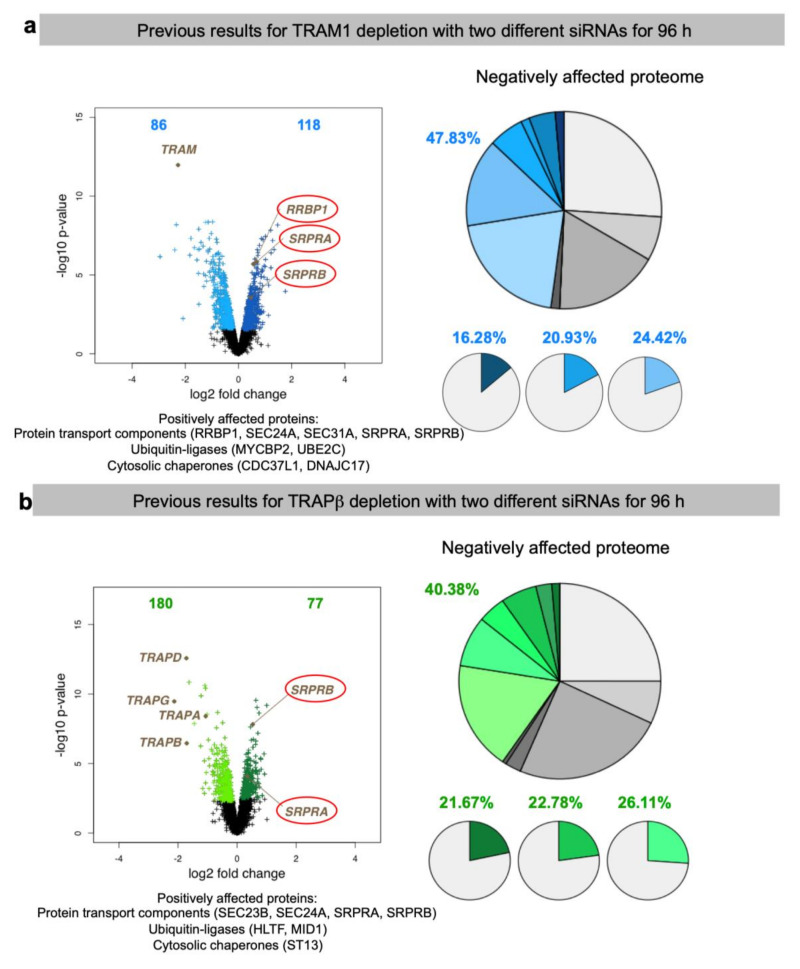
Representative volcano plots and GO enrichment after TRAM1 and TRAP depletion, respectively. Differentially affected proteins were characterized by the mean difference of their intensities plotted against the respective permutation false discovery rate-adjusted *p*-values in volcano plots. The representative results for a single targeting siRNA are shown on the left. *TRAM1* (**a**) and *TRAPB* (**b**), which is synonymous to *SSR2* and codes for TRAPß, *RRBP1* and subunits of the SRP receptor are highlighted in the plots. In addition, the numbers of proteins, which were negatively or positively affected in both experiments, are given; the full set of relevant positively affected proteins are given below. Classification of the respective clients was based on GO enrichment factors where the results from the complete set of quantified proteins (not shown) are compared with the negatively affected proteome. Protein annotations of signal peptides, membrane location, and N-glycosylation in humans were extracted from UniProtKB, and used to determine the enrichment of GO annotations among the negatively affected proteins. The figure summarizes results from previous experiments and is shown here in modified form for comparison [101,102].

**Figure 11 molecules-26-03591-f011:**
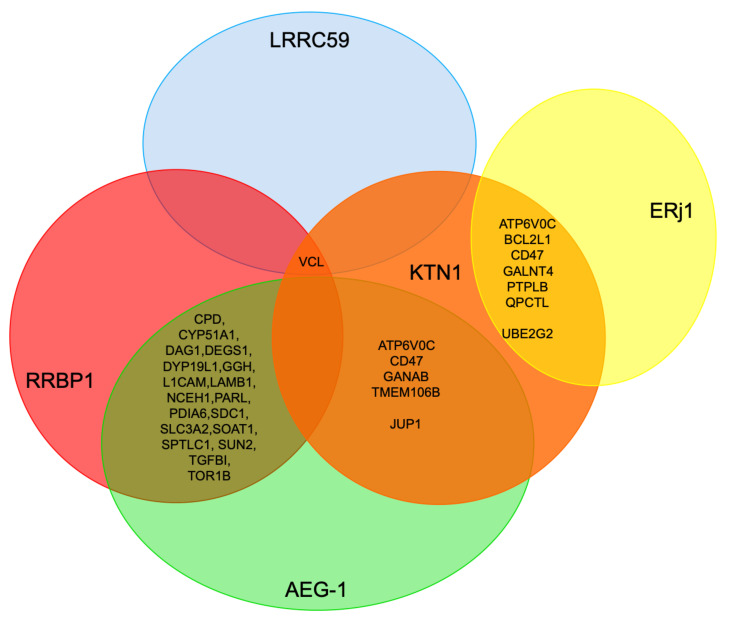
Venn diagram for the overlap of client mRNAs between various mRNA receptors of the ER membrane and ERj1. In addition, COL4A2 was negatively affected by depletion of RRBP1 and KTN1.

**Figure 12 molecules-26-03591-f012:**
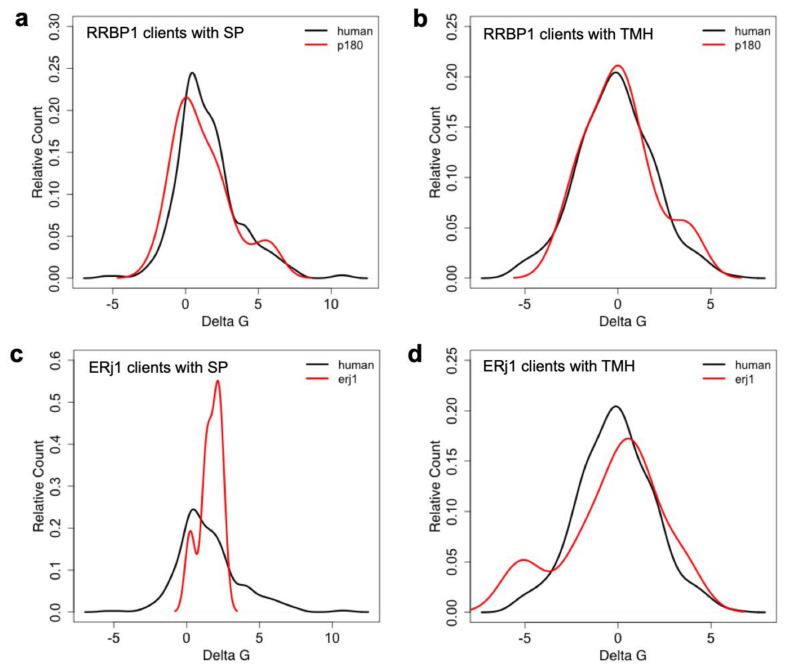
Physico-chemical properties of TRAP clients with SP or TMH. delta G_app_ values of SPs (**a**,**c**) and TMHs (**b**,**d**) were determined for RRBP1 (p180) and ER1 (erj1) clients with the delta G_app_ predictor for TM helix insertion (https://dgpred.cbr.su.se/index.php?p=home, (accessed on 2 May 2021)) and plotted against the relative count in comparison to all human SPs and TMHs.

**Table 1 molecules-26-03591-t001:** Protein targeting and transport components/*complexes* in HeLa cells.

Component/Subunit	AREs	Abundance	Location	Linked Diseases
# p34 (LRC59, LRRC59)	1	2480	ERM	
# p180 (RRBP1)	0	135	ERM	Hepatocellular Carcinoma, Colorectal Cancer
Kinectin 1 (KTN1)	3	263	ERM	
AEG-1 (LYRIC, MTDH)	11	575	ERM	
*# SRP*			C	
- SRP68	1	197		
- SRP54	0	228		Neutropenia, Pancreas Insufficiency
- SRP19	8	33		
- SRP14	0	4295		
- SRP9	12	3436		
-SRP72	5	355		Aplasia, Myelodysplasia
- 7SL RNA				
*SRP receptor*			ERM	
- SRα	1	249		
- SRβ	1	173		
hSnd1		unknown		
*hSnd receptor*			ERM	
- hSnd2 (TMEM208)	0	81		
- hSnd3	1	49		
*# Bag6 complex*			C	
- TRC35 (Get4)	2	171		
- Ubl4A	NA	177		
- Bag6 (Bat3)	1	133	C C ERM	
SGTA	2	549		
TRC40 (Asna1, Get3)	0	381		
*TA receptor*				
- CAML (CAMLG, Get2)	3	5		
- WRB (CHD5, Get1)	4	4		Congenital Heart Disease
PEX1 9PEX3	2 3	80 103	C ERM	Zellweger Syndrome Zellweger Syndrome
*# Sec61 complex*			ERM	
- Sec61α1	1	139		CVID, TKD, Neutropenia
- Sec61β	1	456		PCLD, Colorectal cancer
- Sec61γ	0	400		GBM, Hepatocellular carcinoma
*Sec62/Sec63 complex* # Sec62 (TLOC1) Sec63	14 13	26 168	ERM	Breast-, Prostate-, Cervix-, Lung-Cancer et al. PLD, Colorectal cancer et al.
# ERj1 (DNAJC1)	0	8	ERM	
# TRAM1	6	26	ERM	
TRAM2	3	40	ERM	
*# TRAP complex*			ERM	
- TRAPα (SSR1)	21	568		
- TRAPβ (SSR2)	0			
- TRAPγ (SSR3)	7	1701		CDG, Hepatocellular Carcinoma
- TRAPδ (SSR4)	NA	3212		CDG

Alternative names of components/subunits are given in parentheses. AREs, no. of AU-rich elements in 3′UTR; C, cytosol; CDG, Congenital disorder of glycosylation; CVID, Common variable immunodeficiency; ERM, ER membrane; GBM, Glioblastoma multiforme; NA, not available in AREsite 2 database; PCLD, Polycystic liver disease; TKD, Tubulo-interstitial kidney disease #, ribosome associated.

**Table 2 molecules-26-03591-t002:** Statistics for the identification of RRBP1, KTN1, and ERj1 clients in comparison to Sec61 clients in HeLa cells.

Proteins	SEC61A1	RRBP1	KTN1	ERj1
Quantified proteins	7212	4813	4947	4947
Statistically analyzed proteins	5129	4813	4947	4947
Representing the secretory pathway (%)	26	26	27	27
Proteins with SP (%)	6	6	6	6
N-Glycoproteins (%)	8	8	8	8
Membrane proteins (%)	12	12	13	13
Positively affected proteins	342	157	25	80
Negatively affected proteins	482	141	45	92
Representing the secretory pathway (%)	61	37	41	30
Negatively affected proteins with SP (%)	41	18	7	8
Negatively affected N-glycoproteins (%)	45	17	18	13
Negatively affected membrane proteins (%)	36	18	22	11
Negatively affected proteins with SP	197	21	3	7
Including N-glycoproteins	158	16	3	7
Corresponding to %	80	76	100	100
Including membrane proteins	77	6	1	2
Corresponding to %	39	29	33	29
Negatively affected proteins with TMH	98	18	8	8
Including N-glycoproteins	56	7	4	3
Corresponding to %	57	39	50	38

**Table 3 molecules-26-03591-t003:** Statistics for the identification of BiP, Sec62, Sec63, TRAM1, and TRAP clients in HeLa cells.

Proteins	BiP	SEC62	SEC63	TRAM1	TRAP
Quantified proteins	5543	6686	6655	7502	7670
Statistically analyzed proteins	5543	4819	6655	5961	5911
Representing the secretory pathway (%)	28	28	28	28	27
Proteins with SP (%)	7	7	7	7	7
N-Glycoproteins (%)	9	9	9	9	8
Membrane proteins (%)	14	14	14	14	13
Positively affected proteins	406	196	13	118	77
Negatively affected proteins	340	155	21	86	180
Representing the secretory pathway (%)	28	25	50	48	40
Negatively affected proteins with SP (%)	10	12	14	16	22
Negatively affected N-glycoproteins (%)	11	13	24	21	23
Negatively affected membrane proteins (%)	13	8	38	24	26
Negatively affected proteins with SP	33	18	3	13	38
Including N-glycoproteins	25	12	2	7	28
Corresponding to %	76	67	67	54	74
Including membrane proteins	15	2	2	4	19
Corresponding to %	45	11	67	31	50
Negatively affected proteins with TMH	22	6	6	17	22
Including N-glycoproteins	9	5	3	9	11
Corresponding to %	41	83	50	53	50

## Data Availability

Protein abundances in HeLa cells, given in Table 1, were reported by Hein et al. [132]. The novel mass spectrometry proteomics data have been deposited to the ProteomeXchange Consortium via the PRIDE partner repository with the dataset identifiers: PXD011989 and PXD011990 (http://www.proteomexchange.org, (accessed on 2 May 2021)). In addition, all data are available from the authors.

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
