# Peer review of "Quantitative Proteomics and Differential Protein Abundance Analysis after Depletion of Putative mRNA Receptors in the ER Membrane of Human Cells Identifies Novel Aspects of mRNA Targeting to the ER"

_molecules, 2021, doi:10.3390/molecules26123591_

Round 1
Reviewer 1 Report
The manuscript submitted by Bhadra et al. presents a very elegant method for studying mRNA targetting to the ER using siRNA and mass spec. In this study following depletion, a number of dysregulated proteins were identified, which were reasoned to indicate a relation to a specific pathway of mRNa targetting.
Minor polishing of the English language is required including changes such as:
Line 28 "the question which" to "the question of which"
Line 83 "In case" to "In the case"
Line 334 "Ststistics" to " Statistics"
Line 338 "isoforms, is" to "isoforms, and is"
Line 411 "indication for activation" to "indication of activation"
Line 504 "cell biologically confirmed" to "biologically confirmed"
Line 952 The first part of the sentence starting "KTN1 may be" seems to be missing something in order to fit in with the second part which starts "takes over mRNAs"
In relation to the validation of the data obtained from mass spectrometry, have any Western blots been performed on at least one negatively regulated proteins for each depletion (in the Supplementary Tables) to validate the MS results? It would be worth presenting by how much the data obtained by the MS analysis varies from the Western blot outcome.
Has it been considered during analysis that some of the dysregulated proteins could in effect be fulfilling other roles related to the targetted pathway? How has this been accounted for in the incorporated in the Results analysis presented?
Author Response
Thank you for your encouraging comments on our manuscript and for your help in improving it.
We corrected all minor problems as suggested.
We have included additional validation results from western blots as new Figure 4. In the case of RRBP1 depletion, we have validated the MS results for four proteins, one negatively affected (RRBP1), one positively affected (SRa), and two unaffected proteins (SRP54, Sec62) and confirmed the MS results. Thus four out of four tested proteins from MS results were confirmed by westen blot. In the case of KTN1 depletion, two of two tested proteins from MS results were confirmed. We have added statements along these lines to the manuscript (pages 14,15).
Indeed, we have found earlier that depletion of Sec61a or of TRAPß causes depletion of all subunits of the respective complex, which we attributed to degradation of the additional subunits in response to the absence of a subunit crucial for complex stability. Thus, you are right and we have added statements towards this end to the manuscript (pages 9,11).
Reviewer 2 Report
Bhadra et al. provide in their manuscript a substantial amount of proteomics data on the involvement of various components of the ER targeting machinery.
My impression is that the authors stuffed a number of experiments into this manuscript without going into molecular details to elucidate e.g. one of the pathways with further experiments. Whereas the sheer amout of data is overwhelming, it also stays superficial.
Another flaw is, that all data was generated from Hela cells - limiting the possibility to draw conclusions for healthy cells'/in a physiological context.
In view that this manuscript was submitted for a special issue on proteomics advancements, I will recommend to accept it after "major revision" - mainly because in some Figures the essential controls for knock-down efficiencies are missing.
Further specific comments:
Line 334: Two typos in Table 2 title: "Statistics" and "Sec63"
Line 380: To my knowledge SUN2 is a nuclear envelope- not an ER protein.
Fig.4b: I would like to see at least one molecular weight marker next to the blots. The bActin "band" looks very odd - there is no separation between the lanes? Could the authors provide a better exposure? Also brightness/contrast was adjusted differently for the two blots - this should be corrected as well.
Fig. 5b, Fig. 6b: As for Fig. 4b, mol.weight markers are missing. the "quantification" of the knock-down efficiency does not look reliable to me, eg. ERj1 #siRNA2, sample 1 should be 14% of control? I would suggest to delete the quantifications as the knockdown is anyway clearly visible by eye.
Fig. 7-9: The data confirming knock-down efficiency are missing. New data is presented in this article, hence the proper controls have to be included.
Line 852, 853: There are type-set errors "@elta"
Line 993: The authors replace their HeLa cells only every 5 years? If so, the number of passages the cells are kept in culture before thawing a new vial would be more informative.
Line 1011: List the used antibodies (company, clone or catalog number). The current method description is not acceptable.
Line 1073, 1074: Type-set errors.
Author Response
Thank you for your help in improving our manuscript. However, we are convinced that you went too far in parts of your negative assessment.
You are rigth that the manuscript does not go into the molecular details of either one of the pathways and, thus, stays somewhat superficial, granted. However, we are convinced that it is worthwhile to make these data available to the scientific community anyhow. The reasons are that i) the new data sets add to the data sets on two additional mRNA receptors from C. Nicchitta´s group, on five translocation components from our group, and on four precursor receptor systems from our group (SRP, GET, SND, PEX3/19; manuscript in preparation), and ii) there are other groups interested in further analyzing these data sets (we were already contacted by some colleagues for further details). Having said this, we have included additional validation results from western blots as new Figure 4 (as suggested by reviewer 1). In the case of RRBP1 depletion, we have validated the MS results for four proteins, one negatively affected (RRBP1), one positively affected (SRa), and two unaffected proteins (SRP54, Sec62) and confirmed the MS results. Thus four out of four tested proteins from MS results were confirmed by westen blot. In the case of KTN1 depletion, two of two tested proteins from MS results were confirmed. We have added statements along these lines to the manuscript (pages 14,15).
Although we realize that HeLa cells are derived from a cervix carcinoma, this is the first time that using HeLa cells is considered a flaw in one of our studies. The thing is that there are very solid quantitative MS data on this cell line from M. Mann´s group and, so far, there were no reports that there are odd protein transport pathways in these cells, which are due to the transformation.
It may have escaped your attention that some figures summarize data from previous papers and, therefore, were labeled as such. We have added statements to the legends to point this out more explicitly. They are shown as proof of concept and for comparison, respectively (Figures 3, 8-10). Therefore, the essential respective controls were shown in the previous publications and not repeated here.
Typos were corrected as suggested, thank you. You are right about SUN2, we apologize. Unfortunately, we don´t have a better actin blot for this particular experiment (new Figure 5b). The markers were added and are better visible in the complete images of the Supplementary Material. We have followed your advice and removed the quantitative data from the blots. We have added a statement, concerning the cell passages we routinely use in experiments (page 29). However, some journals ask for the information, you find superfluous. Therefore, we kept it as well. It must have escaped your attention that the antibodies were described in detail in the Materials section of the original manuscript (page 29). As a matter of fact, we even indicated the used dilutions.
Round 2
Reviewer 2 Report
The authors have improved their manuscript and it is acceptable for publication in its present form.
Just as a final remark from my side: HeLa cells (as most other tumor-derived cells) have numerous deregulated pathways and I would be surprised if metabolism would not be affected in some way. I appreciate that for technical reasons many labs use tumor cell lines, but would encourage to confirm data in a more physiological model system in the future.
I apologize that I missed the information about antibodies in the materials section, of course it was appropriately described there already.